# MODEL THEFT AND INVERSION ATTACKS AGAINST QUERY-FREE COLLABORATIVE INFERENCE SYSTEMS

## ABSTRACT

Collaborative inference systems are designed to deploy high-performance models on resource-constrained edge devices by splitting the model into two parts, deployed separately on the client device and the server. However, server-side adversaries can still infer client's private information from the latter part of the model. Previous works rely on auxiliary data with matching labels and unlimited queries to reconstruct inference data or determine sample membership. In contrast, this paper introduces a novel threat called Model Theft and Inversion Attacks (MTIA), targeting a more realistic and challenging scenario where adversaries often lack access to label-consistent datasets. Moreover, adversaries cannot query the client device and have no knowledge of the client model's architecture or parameters. To address these challenges, we leverage transfer learning and self-attention alignment to extract knowledge from the server model and align it with the target task. This enables model recovery with performance comparable to the original model while improving the reconstruction of high-fidelity private data. Additionally, we propose an enhancement that uses reconstructed images to further boost the recovered model's performance. Extensive experiments across various datasets and settings validate the effectiveness, robustness, and generalizability of our approach.

## 1 INTRODUCTION

With the advancement of Deep Neural Networks (DNN), an increasing number of models have been deployed in various edge devices (Tekin et al., 2024), including healthcare (Yang et al., 2021), autonomous driving (Wang et al., 2024) and biometric recognition (e.g., face, fingerprint, and palmprint) (Sardar et al., 2024). Due to the limited computational resources of edge devices, pruned or lightweight models are typically deployed. While this approach reduces computational burden, it also limits the deployment of higher-accuracy models and the ability to handle more complex tasks. To address this, researchers have proposed deploying models in a distributed manner across the server and client devices (Vepakomma et al., 2018), which is called Collaborative Inference (CI) (Li et al., 2018; Kang et al., 2017; Banitalebi-Dehkordi et al., 2021; Li et al., 2021). A common approach is to partition the model into two parts: the front layers are deployed on client devices for feature extraction, while the latter layers reside on the server for further processing. This strategy not only enables the deployment of larger models but also reduces computational costs on client devices. Additionally, by keeping data on the device without direct access by the server, it enhances user privacy.

Current studies show that the server can reconstruct inference data from the intermediate features sent by the client. However, they primarily focus on black-box and white-box scenarios (Zhang et al., 2024; Liu et al., 2024; Yang et al., 2022; Li et al., 2023), where server attackers are typically assumed to have unlimited query access to the client model or full access to its parameters. They largely overlook a more common and realistic yet challenging scenario known as the query-free setting (He et al., 2019; Chen et al., 2020), in which they are unable to perform effective attacks.

In practice, model inputs on the edge device are generated by offline users. For instance, if the front model is used within a company and the server is managed by a third party (e.g., Google Cloud), the service is accessible only to company employees. Since the server is not a valid user and lacks physical access to the client, it cannot send arbitrary queries to the client model. The

query-free setting is more challenging, as the server has access only to the latter half of the model and possesses no knowledge of the client model's architecture or parameters. Furthermore, in face recognition tasks, the server adversary lacks access to a dataset with the same labels. Obtaining such a dataset would be equivalent to acquiring detailed facial data of individuals, which is unrealistic and raises ethical concerns. Since previous attacks rely on auxiliary data with matching labels, the label inconsistency and query-free constraints limit their applicability.

To investigate privacy leakage under the challenging constraints of label inconsistency and query-free setting, this paper proposes a new threat called Model Theft and Inversion Attacks (MTIA), which aims to achieve two goals: recovering the client model functionality and reconstructing the private training data. We first design a transfer-based method to extract hidden knowledge from the server model and reconstruct the missing client model using a label-inconsistent auxiliary dataset. Although the recovered model performs well on the auxiliary dataset, it initially lacks proper alignment with the target task. To address this, we introduce a bottom-up, layer-wise self-attention alignment strategy, enabling the front layers to adapt to and align with the latter layers via attention maps. As a result, the recovered model achieves performance comparable to the full target model, leading to severe model leakage. Empowered by successful model recovery, we introduce a more severe threat—model inversion—to reconstruct private training data. We further propose an enhancement strategy that leverages the reconstructed images as a substitute for the private dataset to fine-tune the recovered model. We evaluate MTIA on two widely used facial datasets, CelebA and FaceScrub, using different models. Experimental results show that MTIA significantly enhances information extraction from deeper layers, improving model recovery performance from 0.13% to 77.05% on CelebA and raising image reconstruction success from 4.13% to 84.79%. Extensive ablation studies across model architectures, dataset sizes, defenses, and datasets further underscore the effectiveness, robustness, and generalizability of our approach. Our contributions are as follows:

- We propose the first Model Theft and Inversion Attacks (MTIA) against collaborative inference systems under label inconsistency and query-free settings, which are both more realistic and challenging.

- We apply a two-step recovery method based on transfer learning and self-attention alignment, which extracts hidden information from deeper layers and achieves better alignment for target tasks. This recovery significantly boosts the success of high-fidelity identity revelation.

- We conduct comprehensive experiments and ablation studies to demonstrate that MTIA achieves remarkable performance across various settings and datasets, further highlighting its robustness and generalizability.

## 2 BACKGROUND AND RELATED WORK

### 2.1 COLLABORATIVE INFERENCE SYSTEMS

Machine learning models require substantial computational resources, making deployment on resource-constrained edge devices challenging. To address this, a paradigm called Collaborative Inference (CI) has been proposed (Li et al., 2018; Kang et al., 2017; Banitalebi-Dehkordi et al., 2021; Li et al., 2021). CI splits the model into two parts: one deployed on the client device and the other on a cloud server. The client model processes raw data and transmits intermediate features to the server for further computation. This setup significantly reduces the computational burden on the client, enabling the use of more powerful models in constrained environments. A related paradigm, Split Learning (Vepakomma et al., 2018), is designed for collaborative training.

Although CI keeps user input local, the server can still infer private information from the server model (Zhang et al., 2024; Liu et al., 2024; Yang et al., 2022; Li et al., 2023; He et al., 2019; Chen et al., 2020). Chen et al. (Chen et al., 2020) and Zhang et al. (Zhang et al., 2024) focus on inferring whether a sample belongs to the training data, while Yang et al. (Yang et al., 2022) and Li et al. (Li et al., 2023) aim to reconstruct the inference data from intermediate features. In contrast to these works, this paper investigates new threats—model theft and inversion attacks—under a more realistic setting where the server faces label inconsistency and operates without the ability to query the client.

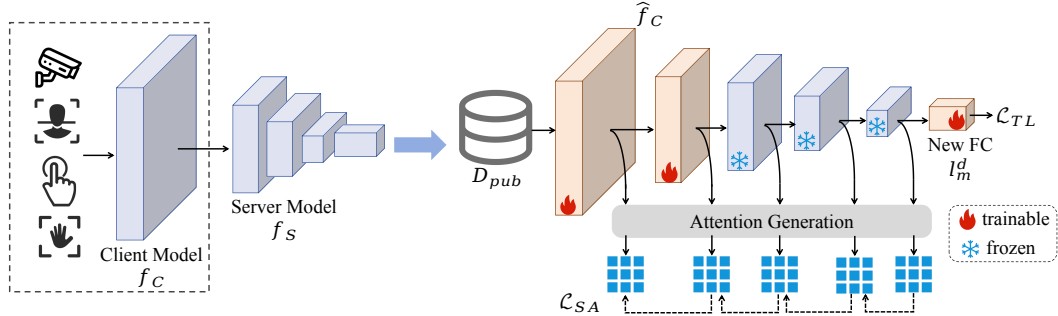

(a) Self-attention guided model recovery

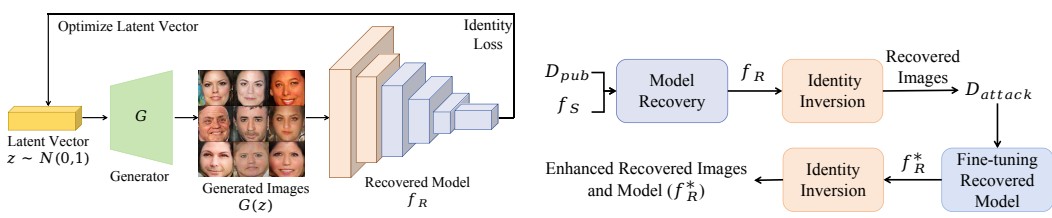

(b) Model inversion attacks on recovered target model        (c) Enhancing attacks using stolen identities

Figure 1: Overview of model theft and inversion attacks.

## 2.2 MODEL INVERSION ATTACKS

Model Inversion Attacks (MIAs) (Zhang et al., 2020; Yuan et al., 2023; Struppek et al., 2022; Qiu et al., 2024) pose a particularly severe risk to training data privacy, aiming to reconstruct it by solving an optimization problem in the input space $x$:

$$x^* = \arg\min_x \ \mathcal{L}_{cls}(T(x), c), \tag{1}$$

where $c$ represents the target class, $T$ denotes the target model, and $\mathcal{L}_{cls}$ refers to the classification loss (e.g., cross-entropy). By minimizing the loss, the input $x$ is optimized to resemble the training data associated with label $c$.

## 3 METHOD

### 3.1 THREAT MODEL

**Attack Scenario.** Given a target model $f_T = \{f_C, f_S\}$ trained on private data $D_{priv} = \{X_{priv}, Y_{priv}, c\}$, where $c$ represents the number of classes, our attack targets model deployed in a distributed manner across a client device ($f_C$) and a server ($f_S$). The server holds the latter $m$ layers, $f_S = \{l_1, ..., l_{m-1}, l_m^c\}$, where the final layer is a fully connected layer producing $c$ class outputs. In this setup, the client processes user inputs $x$ into intermediate features $f_C(x)$, which are then uploaded to the server. The server further processes these intermediate features to generate the final output $f_S(f_C(x))$, which is returned to the client. Crucially, model inputs are controlled by real-world users, ensuring that the server cannot directly input data into the client. For example, consider a scenario where the client is deployed within a company and the server is managed by a third party. In this case, the server cannot upload data to the client model, as it is used exclusively by company staff. This scenario is more realistic and presents a greater challenge.

**Adversary's Goal.** We consider an adversary, either an external entity that compromises the server or the server itself. The adversary has two primary objectives: (1) Model Leakage – Recover the client model's functionality to reach performance close to that of the full target model $f_T$. (2) Data Leakage – Reconstruct the private training images utilized in the target model.

**Adversary's Knowledge.** We consider a practical and realistic scenario where the adversary has access only to the server model and a public dataset $D_{pub} = \{X_{pub}, Y_{pub}, d\}$ containing $d$ classes from the same domain as the training dataset. Importantly, the public dataset has no class overlap with the target training dataset, and both the number of classes ($c \neq d$) and their distribution differ. This constraint prevents the adversary from directly leveraging the public dataset to reconstruct the complete model. Furthermore, the adversary has no knowledge of the client model's weights or architecture and cannot query the client.

## 3.2 SELF-ATTENTION GUIDED MODEL RECOVERY

Since $D_{pub}$ has a different distribution from $D_{priv}$, directly training a new model on $D_{pub}$ or combining it with $f_S$ is ineffective, as the training tasks differ. To address this, we propose a model recovery method based on transfer learning and self-attention alignment to fully extract the knowledge embedded in $f_S$. The workflow as shown in Figure 1 (a), consists of the following two steps:

**Step1: Transfer-based Model Completion.** To fully leverage the information hidden in the deeper layers, we first completes the model using a weight transfer learning approach. We initialize a new feature extractor, $\hat{f}_C$, as a substitute for $f_C$. Since the adversary lacks knowledge of the model's architecture, the architecture of $\hat{f}_C$ differs from that of $f_C$. However, $\hat{f}_C$ cannot be directly combined with $f_S$ because the last layer $l_m^c$ in $f_S$ is mismatched with the class numbers of $D_{pub}$. To address this, a new classification layer $l_m^d$ is initialized, replacing the last layer in $f_S$ to form a new public model: $f_P = \{\hat{f}_C, \hat{f}_S = \{l_1, ..., l_{m-1}, l_m^d\}\}$. The public model is then trained on $D_{pub}$:

$$\mathcal{L}_{TL} = \mathcal{L}_{cls}(\hat{f}_S(\hat{f}_C(X_{pub})), Y_{pub}) \tag{2}$$

where $\mathcal{L}_{cls}$ denotes the classification loss. The layers from the server model $\{l_1, ..., l_{m-1}\}$ remain frozen, and only the newly initialized layers, $\hat{f}_C$ and $l_m^d$, are trained.

**Step2: Model Fine-tuning using Self-attention Alignment.** Although we can complete the model using $D_{pub}$, the feature extractor is trained specifically for $D_{pub}$, resulting in poor performance on $D_{priv}$. To better align the feature extractor with the target's deeper layers and tasks, we adopt Self-Attention Distillation (SAD) (Hou et al., 2019). SAD was originally proposed to enhance the model's representation learning through no teacher distillation; in this work, we adopt it as an alignment mechanism, allowing the front layers to learn from the latter layers through attention maps.

The attention maps are obtained by processing the output of a specific layer. We denote the output of a layer as $A \in \mathbb{R}^{C \times H \times W}$, where $C$, $H$, and $W$ represent the channel, height, and width, respectively. To construct an attention mapping function, we define $\mathcal{G}(A) = \sum_{j=1}^{C} |A^j|^2$. This function is derived by computing statistical properties across the channel dimension. The absolute value of each element in the resulting attention map indicates its importance in determining the final output. We conduct a bottom-up, layer-wise alignment that utilizes the attention maps of the deeper layers as supervision for the shallower layers. The self-attention alignment loss, $\mathcal{L}_{SA}$, is defined as:

$$\mathcal{L}_{SA}(A_i, A_{i+1}) = \|\Phi\left(\mathcal{U}\left(\mathcal{G}(A_i)\right)\right) - \Phi\left(\mathcal{U}\left(\mathcal{G}(A_{i+1})\right)\right)\|_2 \tag{3}$$

$$\mathcal{G}(A) = \sum_{j=1}^{C} |A^j|^2, \quad \Phi(A) = \frac{\exp(A)}{\sum_{h,w} \exp(A_{h,w})} \tag{4}$$

$$\mathcal{U}(A)_{u,v} = \sum_{i=1}^{H} \sum_{j=1}^{W} A_{i,j} \cdot \max(0, 1 - |u' - i|) \cdot \max(0, 1 - |v' - j|) \tag{5}$$

$$u' = \frac{u \cdot H}{H'}, \quad v' = \frac{v \cdot W}{W'} \tag{6}$$

where $A_i$ denotes the output of the $i$-th layer. $\mathcal{G}$ is an attention mapping function. $\mathcal{U}$ denotes the bilinear upsampling operation that resizes the attention map to a predefined resolution $H' \times W'$. The upsampling is computed based on bilinear interpolation weights derived from the relative positions $u'$ and $v'$ mapped to the original spatial coordinates. $\Phi$ is a spatial softmax function applied over the spatial domain to normalize the attention map into a probability distribution.

The self-attention alignment loss $\mathcal{L}_{SA}$ is computed between consecutive layers and from the penultimate layer's output to the first layer. After alignment, the adversary replaces the last layer $l_m^d$ in $f_P = \{\hat{f}_C, \hat{f}_S = \{l_1, ..., l_{m-1}, l_m^d\}\}$ with the original classification layer $l_m^c$, forming the final recovered target model: $f_R = \{\hat{f}_C, f_S = \{l_1, ..., l_{m-1}, l_m^c\}\}$.

### 3.3 INVERSION-BASED IDENTITY RECONSTRUCTION

After recovering the target model, which performs well on the private data $D_{priv}$, we can apply white-box MIAs on the recovered model $f_R$ to reveal private training data. The attack workflow is shown in Figure 1 (b). The adversary can leverage the public dataset $D_{pub}$ to train a GAN (Yuan et al., 2023) or utilize a pretrained StyleGAN (Struppek et al., 2022) for the attack. The images generated by the GAN are denoted as $G(z)$, where $z \sim \mathcal{N}(0, 1)$ represents the latent vector. The optimization process is formulated as:

$$z^* = \arg\min_z \mathcal{L}_{cls}(f_R(G(z)), y_t), \tag{7}$$

where $y_t$ is the target class, $f_R$ is the recovered model, and $\mathcal{L}_{cls}$ denotes the classification loss (identity loss). The reconstructed images can be obtained as $x^* = G(z^*)$.

### 3.4 ENHANCING ATTACKS USING STOLEN IDENTITIES

After the inversion attacks, the adversary can obtain images for each identity. These reconstructed images can be viewed as an approximate substitute for $D_{priv}$. Consequently, the adversary can create a new dataset $D_{attack}$ using the reconstructed images and employ it to fine-tune the recovered model, thereby enhancing its performance on $D_{priv}$. This process is referred to as *repeated attack*. The process of repeated MTIA (r-MTIA) is illustrated in Figure 1 (c). After obtaining an enhanced recovered model $f_R^*$, the adversary can once again reconstruct the private training data.

## 4 EXPERIMENTS

### 4.1 EXPERIMENTAL SETUP

**Datasets.** We choose the face classification task for our main experiments and utilize two widely used datasets: CelebA (Liu et al., 2015) and FaceScrub (Ng & Winkler, 2014). CelebA consists of 202,599 face images from 10,177 identities. For our experiments, we select 30,027 images from 1,000 identities. FaceScrub contains 106,863 images of 530 individuals. We use the entire FaceScrub dataset. For both datasets, when one is designated as $D_{priv}$, the other serves as $D_{pub}$, ensuring that $D_{priv}$ and $D_{pub}$ are distributed differently and have no overlap. All images are resized to $224 \times 224$. We also conduct experiments on other datasets, more details can be found in the Appendix D.

**Models.** We employ two different model architectures: MobileNetV2 (Sandler et al., 2018) and ResNet-50 (He et al., 2016). Since the adversary lacks knowledge of the complete model architectures, we use VGG blocks (consisting of two convolutional layers) (Simonyan, 2014). Notably, VGG blocks differ significantly from the residual blocks in ResNet-50 and the inverted residual blocks in MobileNetV2 in terms of channel dimensions and the number of layers. Refer to the Appendix E for detailed architectural differences.

**Attacks and Defenses.** We select two white-box MIAs: PLGMI (Yuan et al., 2023) and PPA (Struppek et al., 2022). PLGMI utilizes a self-trained GAN, while PPA employs a StyleGAN pretrained on the FFHQ dataset (Karras et al., 2019). Additionally, we select three MIA defenses and five collaborative inference defenses for evaluation: BiDO (Peng et al., 2022), NLS (Struppek et al., 2023), and TLDMI (Ho et al., 2024) for MIAs, and NoPeek (Vepakomma et al., 2020), Noise (Titcombe et al., 2021), Dropout (He et al., 2020), DISCO (Singh et al., 2021), and InfoScissors (Duan et al., 2024) for CI. Moreover, we also include Differential Privacy (DP) (Abadi et al., 2016), a widely used method for privacy protection. See Appendix C for more details and hyperparameters.

**Evaluation Metrics.** To evaluate the attack performance, we conducted both qualitative evaluation through visual inspection and quantitative evaluation using three metrics:

Table 1: Attack results across various datasets and models. ↑ and ↓ indicate that higher and lower scores, respectively, correspond to better attack performance.

| Dataset | Method | MobileNetV2 | | | | | ResNet-50 | | | | |
| | | TestAcc ↑ | PLGMI | | PPA | | TestAcc ↑ | PLGMI | | PPA | |
| | | | AttAcc ↑ | FDist ↓ | AttAcc ↑ | FDist ↓ | | AttAcc ↑ | FDist ↓ | AttAcc ↑ | FDist ↓ |
| CelebA | Target | 88.15 | 87.06±1.4 | 174.30 | 90.93±1.3 | 122.56 | 87.67 | 84.53±1.2 | 184.00 | 92.33±0.8 | 150.23 |
| | Pretrain | 0.06 | 0.66±0.2 | 240.62 | 0.00±0.1 | 306.29 | 0.09 | 0.26±0.3 | 228.52 | 0.06±0.1 | 306.81 |
| | Pretrain-SA | 8.95 | 29.33±1.0 | 213.21 | 10.13±0.1 | 226.79 | 0.87 | 0.66±0.2 | 205.89 | 0.73±0.4 | 263.84 |
| | TL | 0.13 | 4.13±1.2 | 195.36 | 0.86±0.5 | 293.12 | 33.39 | 66.60±1.4 | 196.75 | 60.46±3.0 | 202.37 |
| | MTIA | 77.05 | **84.79±1.4** | 177.86 | 72.93±2.2 | 149.47 | 68.32 | 63.53±0.8 | **192.21** | 65.73±1.3 | 183.87 |
| | r-MTIA | **79.62** | 81.46±1.1 | **173.98** | **77.46±2.2** | **141.18** | **71.38** | **67.26±0.4** | 199.04 | **70.26±2.7** | **181.59** |
| FaceScrub | Target | 93.48 | 97.06±0.8 | 142.47 | 94.66±1.1 | 121.85 | 93.96 | 94.59±1.6 | 143.01 | 96.33±0.5 | 136.68 |
| | Pretrain | 0.02 | 1.99±0.4 | 197.59 | 0.20±0.2 | 269.66 | 0.18 | 0.26±0.3 | 202.56 | 0.13±0.1 | 295.18 |
| | Pretrain-SA | 9.07 | 12.53±1.0 | 197.48 | 5.73±1.4 | 230.19 | 3.72 | 0.40±0.1 | 196.34 | 1.33±0.3 | 253.89 |
| | TL | 0.29 | 20.39±2.7 | 185.00 | 1.26±0.5 | 267.47 | 58.86 | 87.06±1.3 | 167.83 | 78.46±1.9 | 175.95 |
| | MTIA | 88.37 | 93.00±0.8 | 141.31 | 84.86±2.0 | 134.25 | 85.93 | 89.06±1.2 | **153.10** | 83.99±2.0 | 152.03 |
| | r-MTIA | **90.07** | **95.53±0.8** | **126.39** | **89.26±0.9** | **129.89** | **86.55** | **89.46±0.6** | 158.73 | **87.00±1.9** | **150.23** |

- *Test Accuracy (TestAcc).* Test accuracy is used to evaluate the performance of the recovered model on $D_{priv}$. A higher test accuracy indicates greater functional similarity, which suggests a higher degree of model theft.

- *Attack Accuracy (AttAcc).* We employ an evaluation model to classify the reconstructed images, measuring inversion attack accuracy. This model is trained on the same $D_{priv}$ but uses a different architecture. High accuracy indicates a successful attack and potential private information leakage. We use InceptionV3 (Szegedy et al., 2016) as the evaluation model.

- *Feature Distance (FDist).* It is evaluated using the penultimate layer outputs of the evaluation model. We measure the $l_2$ distance between the reconstructed image and the nearest private image with the same label. A lower feature distance indicates a closer semantic similarity.

**Baselines.** As this is the first work addressing model and identity theft in such a limited setting, we evaluate two baselines for comparison:

- *Pretrain:* A new model is trained using $D_{pub}$ without utilizing $f_S$. The front layers of this new model are then combined with $f_S$ to form the recovered model.

- *Transfer Learning (TL)*(He et al., 2019; Chen et al., 2020): The model is completed by freezing $f_S$ and training the remaining layers using $D_{pub}$ (Step1).

We further add self-attention alignment (Step2) to both baselines, yielding Pretrain-SA and MTIA (ours), to highlight the impact of the two key steps. Implementation details and hyperparameters are provided in the Appendix D.

## 4.2 MAIN RESULTS

**Comparison with baselines.** Table 1 presents the MTIA results when the server is missing one block. *Pretrain* fails with near-zero accuracy, and even with self-attention alignment (*Pretrain-SA*), performance remains below 10%. *TL* yields just 0.13% and 0.29% accuracy on MobileNetV2, but improves to 33.39% and 58.86% on ResNet-50, benefiting from deeper layer knowledge. Our method, MTIA, significantly outperforms baselines, achieving 77.05% on CelebA and 88.37% on FaceScrub, which are closer to the target model's 88.15% and 93.48%. This demonstrates the superiority of MTIA's two-step knowledge extraction and alignment. Poor model recovery by the baselines leads to failed image reconstruction. However, MTIA boosts image reconstruction, achieving high attack success rates of 84.79% on CelebA and 93.00% on FaceScrub. The visual comparison in Figure 2 further highlights these results. Baseline methods produce unrecognizable images or entirely incorrect identities, whereas MTIA reconstructs images with greater detail and higher quality. This improvement is attributed to the effective model recovery enabled by MTIA.

**Effectiveness of repeated MTIA.** We collect the reconstructed images from PPA on MTIA as the attack dataset $D_{attack}$ for fine-tuning. As shown in Table 1, r-MTIA further enhances the performance of the recovered model, increasing accuracy from 77.05% to 79.62% on CelebA and from

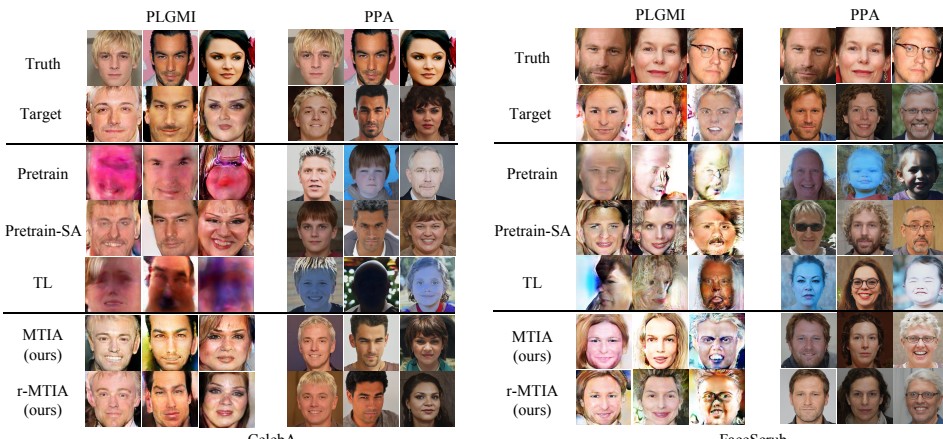

Figure 2: Identity inversion results on MobileNetV2. "Target" refers to the reconstructed images of the target model.

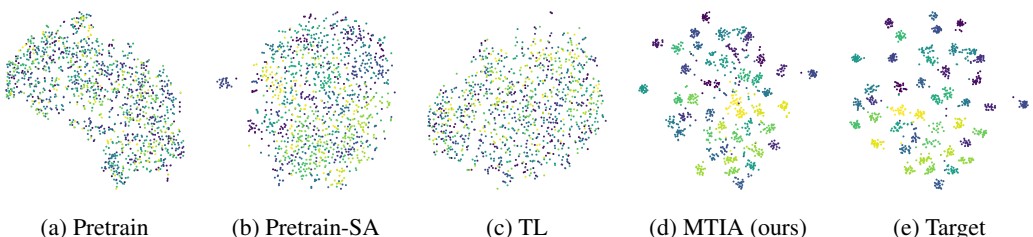

(a) Pretrain      (b) Pretrain-SA      (c) TL      (d) MTIA (ours)      (e) Target

Figure 3: T-SNE visualization on CelebA and MobileNetV2.

88.37% to 90.07% on FaceScrub. Correspondingly, identity revelation attacks also improve. The r-MTIA step can be repeated multiple times, but a single repetition is sufficient. Additional repetitions increase computational cost and time, while yielding diminishing performance gains.

**Effectiveness of two steps in MTIA.** Using only transfer learning (*TL*, Step 1) fails on Mobile-NetV2 and gives moderate results on ResNet-50. Using only self-attention alignment (*Pretrain-SA*, Step 2) provides small improvements but better task alignment. This is evident in Figure 2, where the reconstructions from *Pretrain-SA* show some similarity to the ground truth, particularly in hairstyle and the presence of glasses. When combined (MTIA), model recovery performs best, capturing finer details more accurately.

### 4.3 MODEL SIMILARITY ANALYSIS

To analyze how the model recovered by MTIA behaves like the target model, we use Loss-Rank Correlation (LRC) (Kaya & Dumitras, 2021) to quantify model similarity and t-SNE (Van der Maaten & Hinton, 2008) for feature space visualization. The LRC score is computed as the Spearman's rank correlation coefficient (Spearman, 1904) between the loss values of two models evaluated on the same dataset. The LRC is defined as:

$$\text{LRC} = \frac{\text{cov}(\text{rank}(L_1), \text{rank}(L_2))}{\sigma_{\text{rank}(L_1)} \cdot \sigma_{\text{rank}(L_2)}} \tag{8}$$

where $L_1$ and $L_2$ are the loss vectors from two models, $\text{rank}(\cdot)$ denotes the rank transformation of the loss vector, and cov and $\sigma$ denote covariance and standard deviation.

The LRC score ranges from -1 to 1, with a higher value indicating greater similarity. As shown in Table 2, the model recovered by MTIA achieves a high score across both models, indicating strong similarity at the model level. As shown in Figure 3, the t-SNE plot also reveals greater similarity

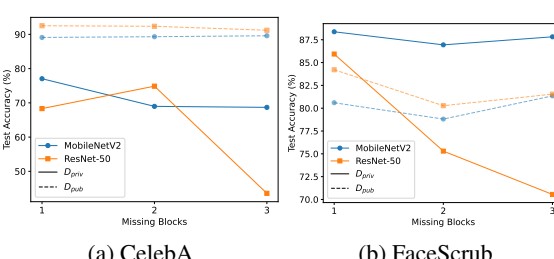

(a) CelebA  (b) FaceScrub

Figure 4: Test accuracy of $f_R$ with different missing blocks.

Figure 5: Test accuracy of $f_R$ on $D_{priv}$ when fine-tuning 20 epochs in Step 2 on MobileNetV2.

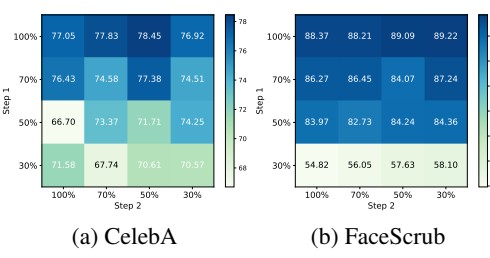

(a) CelebA  (b) FaceScrub

Figure 6: Test accuracy of $f_R$ when using different public dataset size on MobileNetV2.

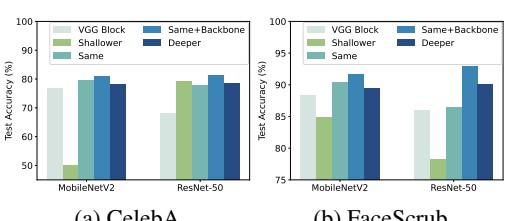

(a) CelebA  (b) FaceScrub

Figure 7: Test accuracy of $f_R$ when using different architectures.

and clearer class boundaries, reflecting closer alignment with the target model at the feature level. The analysis demonstrates the strong success of the model recovered by MTIA.

## 4.4 ABLATION STUDY

**Effect of different missing blocks in $f_S$.** As the server model may have more missing blocks, the server needs to recover a larger portion of the model. To assess performance, we report the model's accuracy on both $D_{priv}$ and $D_{pub}$, replacing the last fully connected layer accordingly. As shown in Figure 4, when more blocks are missing, accuracy on $D_{pub}$ remains nearly unchanged, but the recovered model's performance gradually decreases. This indicates that there is little correlation between performance on $D_{priv}$ and $D_{pub}$. For ResNet-50, the decline is more significant than MobileNetV2. Because a higher number of missing blocks reduces the available information in $f_S$ and increases the architectural differences, making recovery more challenging.

Table 2: LRC score on CelebA.

| Method | MobileNetV2 | ResNet-50 |
|---|---|---|
| Pretrain | -0.161 | -0.360 |
| Pretrain-SA | -0.130 | -0.218 |
| TL | -0.169 | 0.236 |
| MTIA (ours) | **0.831** | **0.738** |

**Effect of fine-tuning epochs.** We further analyze the effect of fine-tuning epochs in self-attention alignment (Step 2). We fine-tune for 20 epochs using MobileNetV2 and report test accuracy on $D_{priv}$ at each iteration, as shown in Figure 5. During fine-tuning, test accuracy gradually improves and reaches its peak. However, with additional fine-tuning, performance begins to fluctuate and slowly declines. Thus, fine-tuning for 5 epochs is sufficient for Step 2.

**Effect of public dataset size for two steps.** We conduct additional experiments using three portions of public dataset: 70%, 50%, and 30% of the original $D_{pub}$, as shown in Figure 6. When the dataset size is reduced for Step 1 of MTIA, the performance of the recovered model slightly decreases. The performance drops to 58.1% for FaceScrub only when using 30% of the dataset, as this limited data makes it challenging to extract meaningful features and achieve accurate classification. For Step 2 of MTIA, reducing the dataset size has nearly no effect on the attack performance. This may be because sufficient hidden information has already been extracted during Step 1, allowing the model's knowledge to be aligned effectively in Step 2.

Table 3: Attack results against MIA defenses.

| Defense | Hyperparams | Method | Test-Acc ↑ | PPA Att-Acc ↑ |
|---|---|---|---|---|
| w/o | - | Target | 88.15 | 90.93±1.3 |
| | | MTIA | 77.05 | 72.93±2.2 |
| BiDO | (0.05, 0.5) | Target | 86.98 | 87.93±0.6 |
| | | MTIA | 71.84 | 70.59±1.4 |
| NLS | -0.005 | Target | 87.31 | 74.59±2.4 |
| | | MTIA | 69.35 | 55.59±3.6 |
| TLDMI | 0.5 | Target | 84.71 | 78.20±1.5 |
| | | MTIA | 66.89 | 55.73±1.2 |
| DP | 0.01 | Target | 83.44 | 81.80±1.8 |
| | | MTIA | 56.57 | 51.39±3.2 |

Table 4: Attack results against CI defenses.

| Defense | Hyperparams | Method | Test-Acc ↑ | PPA Att-Acc ↑ |
|---|---|---|---|---|
| NoPeek | 0.7 | Target | 87.83 | 92.59±1.5 |
| | | MTIA | 77.76 | 79.33±3.5 |
| Noise | 10 | Target | 84.71 | 84.73±1.5 |
| | | MTIA | 72.10 | 77.46±1.7 |
| Dropout | 0.5 | Target | 84.22 | 90.93±1.6 |
| | | MTIA | 75.52 | 73.26±2.8 |
| DISCO | (0.8, 0.5) | Target | 86.85 | 75.59±1.1 |
| | | MTIA | 76.56 | 77.20±2.5 |
| InfoScissors | 0.5 | Target | 88.58 | 91.39±0.7 |
| | | MTIA | 72.00 | 72.13±2.2 |

Table 5: Attack results on other datasets. "Pre" stands for "Pretrain".

| Task | $D_{priv}$ | $D_{pub}$ | Target | Pre | Pre-SA | TL | MTIA |
|---|---|---|---|---|---|---|---|
| Face Classification | CelebA | CelebA (different ID) | 88.15 | 0.03 | 11.26 | 0.13 | **76.80** |
| | CelebA | AI-Face (synthetic) (Lin et al., 2025) | 88.15 | 0.09 | 2.44 | 0.09 | **52.73** |
| Fingerprint Classification | UareU (Neurotechnology, 2007) | FVC2004 (Maltoni et al., 2009) | 98.52 | 1.47 | 17.64 | 1.47 | **71.32** |
| Palmprint Classification | PCE (Jin et al., 2024) | PCE (different ID) | 100.00 | 0.26 | 23.79 | 1.44 | **97.83** |
| Object Classification | Imagenette (Howard, 2019) | Imagewoof | 95.75 | 10.06 | 42.41 | 13.34 | **82.52** |
| | Imagewoof (Howard, 2019) | Imagenette | 87.38 | 5.64 | 21.30 | 5.64 | **51.41** |

**Effect of recovered model architectures.** Since the adversary has access to the server model, they can infer potential model architectures and use the same type of building blocks. We consider three additional architectures using the same type of blocks: with (1) shallower, (2) same, and (3) deeper blocks. For the same architectures, a key point is that frameworks like PyTorch offer popular models with ImageNet-pretrained weights (e.g., *torchvision.models* (PyTorch, 2023; Deng et al., 2009)), which many users adopt as backbones. This means an adversary could download the same pretrained weights as the target model, making recovery easier. As shown in Figure 7, using the same or deeper blocks improves recovery accuracy due to greater similarity to the target model than the VGG block. In contrast, shallower blocks cause notable degradation, as their shallow front layers fail to extract meaningful features. Moreover, knowing the pretrained backbone allows the adversary to achieve the best recovery performance. Our experiments show that architecture choice has limited impact on recovery, as adversaries can compensate with deeper models. More importantly, these findings highlight that using a pretrained backbone increases privacy risks.

**Effect of defenses.** We evaluate MIA and CI defenses on CelebA using MobileNetV2, with results presented in Table 3 and 4. Compared to the model recovery performance on an undefended model (77.05%), these defenses reduce MTIA effectiveness to 56.57% – 77.76%, which in turn further impacts identity reconstruction. However, these defenses are still not highly effective against MTIA, as a model with nearly 60% accuracy can still lead to significant privacy leakage.

**Effect on other dataset.** We test MTIA on various datasets and tasks; more details are provided in the Appendix D. The results are shown in Table 5. For CelebA, even when using data from different identities and a synthetic dataset, MTIA still successfully recovers model functionality. Additionally, for fingerprint, palmprint, and object classification tasks, MTIA achieves attack performance that surpasses the baselines. These results further demonstrate the effectiveness and generalizability of our approach.

**Effect on other models.** We evaluate MTIA on MaxViT (Tu et al., 2022) in Table 13 in Appendix F.9, demonstrating its feasibility on more complex transformer-based architectures.

## 5 CONCLUSION

This paper proposes MTIA, the first model theft and inversion attacks under label inconsistency and query-free settings in collaborative inference. The server adversary uses transfer learning and self-attention alignment to recover the client model, and reconstruct its training data.

ETHICS STATEMENT

All experiments in this work were conducted within controlled research environments. The datasets employed are publicly available, open-source, and were used strictly in accordance with their respective licenses. The proposed attack was never executed on real-world systems; it is studied exclusively for academic and research purposes. Our intention is not to harm or exploit any system or individual but rather to raise awareness of potential privacy risks and to promote the development of stronger privacy-preserving techniques.

REPRODUCIBILITY STATEMENT

All experiments are conducted on a Linux server with CUDA 11.8, Python 3.10, PyTorch 2.0.1, Torchvision 0.15.2, and two NVIDIA GeForce RTX 4090 GPUs. The detailed data processing steps and hyperparameters of MTIA are provided in Appendix D. The hyperparameters of MIA attacks and defenses are listed in Appendix C. The source code is available at `https://anonymous.4open.science/r/MTIA-C2FC`.

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

## A    THE USE OF LARGE LANGUAGE MODELS (LLMS)

We only used LLMs such as ChatGPT for polishing the writing and checking grammar, without employing them for any other purpose.

## B    COLLABORATIVE INFERENCE SYSTEMS

Figure 8 illustrates the framework of Collaborative Inference systems (CI). CI splits the model into two parts: one deployed on the client device and the other on a cloud server. The client model processes raw data and transmits intermediate features to the server for further computation. The server then returns the results to the client user. This setup reduces the computational burden on the client while preserving user privacy. In the query-free setting (He et al., 2019; Chen et al., 2020), the client device is deployed within the company, and the server model is held by a third-party cloud. For example, in a facial recognition attendance system for employees, the system operates offline daily, with the cloud only processing data sent by the client. Consequently, the server cannot query the client, as the client device is located within the company premises.

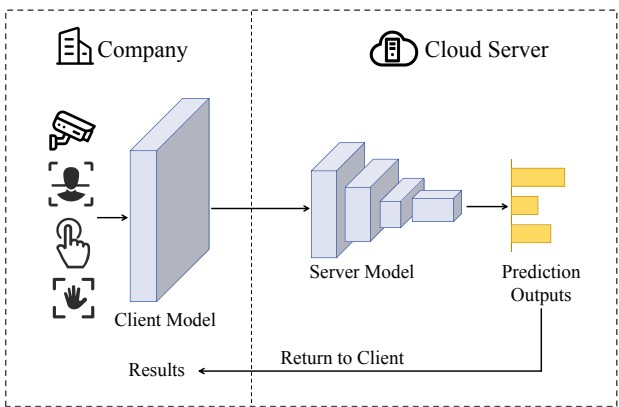

Figure 8: Collaborative Inference Systems.

This setting is more difficult to attack. The client-provided instances are not controllable by the server, which may lead to situations where no instances are sent, such as when the client device is offline or shut down. The attacker cannot freely issue queries and can only passively wait for incoming instances. The attacker does not have access to the original private inputs, which makes the received intermediate features difficult to exploit. In non-query-free methods, the attacker can send auxiliary data to the client and obtain corresponding intermediate features to learn the feature–input mapping or steal the client model through feature distillation. Therefore, in the non-query-free setting, the attacker can access substantially more information. In the query-free setting, the available information is limited, and the instances may also be insufficient.

## C    MODEL INVERSION ATTACKS AND DEFENSES

### C.1    WORKFLOW OF MIAS

Model Inversion Attacks (MIAs) aim to reconstruct sensitive training data from the target model. MIAs can be formulated as an optimization problem in the input space $x$:

$$x^* = \arg\min_x \mathcal{L}_{cls}(T(x), c), \tag{9}$$

where $c$ represents the target class, $T$ denotes the target model, and $\mathcal{L}_{cls}$ refers to the classification loss function (e.g., cross-entropy loss). By minimizing the classification loss of $T$, the input $x$ is optimized to resemble the training data associated with label $c$, potentially revealing sensitive features such as the face of the individual corresponding to class $c$.

With the application of Generative Adversarial Networks (GANs) (Goodfellow et al., 2014; Miyato & Koyama, 2018), the images generated by attackers are no longer random but are instead produced through GANs. Moreover, attackers can leverage components such as the GAN discriminator to further enhance the realism of the generated images. The basic attack workflow is shown in Figure 9. This process can be formally expressed as:

$$z^* = \arg\min_z \; \mathcal{L}_{cls}(T(G(z)), c) + \mathcal{L}_{prior}(G(z)), \tag{10}$$

Where $z$ is the latent code of the GAN, $G$ is the generator, and $\mathcal{L}_{prior}$ represents the loss from the discriminator.

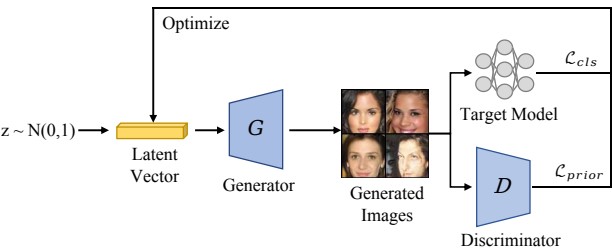

Figure 9: Basic workflow of Model Inversion Attacks.

## C.2 ATTACKS

The first MIA was introduced by Fredrikson *et al.* (Fredrikson et al., 2014; 2015), leveraging gradient descent optimization in the image space. However, the vast size of the image space made optimization challenging, often resulting in unrecognizable reconstructed images. To overcome this limitation, Zhang *et al.* (Zhang et al., 2020) proposed Generative Model Inversion (GMI), which utilizes a GAN to constrain the optimization space and synthesize high-quality reconstructed samples. More recently, advanced variants of GMI have been developed to enhance attack performance under diverse attacker capabilities (Chen et al., 2021; Wang et al., 2021; Struppek et al., 2022; An et al., 2022; Nguyen et al., 2023; Yuan et al., 2023; Qiu et al., 2024; Peng et al., 2024; Han et al., 2023; Li et al., 2025; Kahla et al., 2022; Nguyen et al., 2024).

**PLGMI (Yuan et al., 2023).** PLGMI uses pseudo labels to narrow the search space and conduct a more independent latent search process. PLGMI first selects the best matching $n$ images for each identity from public data, then uses these images to train a conditional GAN (Miyato & Koyama, 2018), which better guides the direction of the generated images. In the attack phase, PLGMI designs a max-margin loss to address the gradient vanishing problem.

*Hyperparameters.* We set $n = 30$ for the top-$n$ selection strategy. To train the GAN, we use the Adam optimizer with a learning rate of 0.0002, a batch size of 64, and $\beta = (0, 0.9)$, training for 150,000 iterations. During the attack, we use the Adam optimizer with a learning rate of 0.1, and initialize $z$ for 5 times and optimize each round for 600 iterations.

**PPA (Struppek et al., 2022).** PPA leverages pre-trained StyleGANs on image priors with large distributional shifts. The attack consists of three phases: sampling, optimization, and selection. In the sampling phase, PPA generates a large pool of latent vectors and selects some vectors for each identity that achieves the highest accuracy on the target model. Then, in the optimization phase, PPA optimizes each latent vector under random transformations and employs a Poincaré loss to address the gradient vanishing problem. Finally, in the selection phase, PPA filters out poor results by evaluating various transformed versions of each corresponding image on the target model and selecting the average best results.

*Hyperparameters.* In the sampling phase, we initially sampled 5,000 latent vectors $z$ and then selected the top 20 candidates with the highest prediction scores. During the optimization phase, we employed the Adam optimizer with a learning rate of 0.005, a batch size of 30, and $\beta = (0.1, 0.1)$, training for 100 iterations. Finally, in the selection phase, we select 5 samples with the highest average prediction scores for each target.

In other settings, attackers may only have access to the model's prediction outputs rather than its full parameters. Based on the type of outputs available, these settings can be further classified into black-box (soft label) (An et al., 2022; Han et al., 2023) and label-only (hard label) scenarios (Kahla et al., 2022; Nguyen et al., 2024). In the black-box setting, MIRROR (An et al., 2022) employs a genetic algorithm for gradient-free optimization, while RLBMI (Han et al., 2023) formulates the latent space search as a Markov decision process and solves it using reinforcement learning. In the label-only setting, BREPMI (Kahla et al., 2022) introduces an algorithm that pushes samples away from the decision boundary and closer to the class centroid. LOKT (Nguyen et al., 2024) transfers knowledge from the target model to surrogate models using hard-label distillation, then performs white-box attacks on the surrogate models.

### C.3 DEFENSES

**BiDO (Peng et al., 2022).** BiDO utilizes a bilateral dependency optimization strategy to minimize the dependency $d(z, x)$ between the latent representations $z$ and the inputs $x$ while maximizing the dependency $d(z, y)$ between the latent representations $z$ and the label $y$. For the dependency measure, BiDO uses constrained covariance (COCO) (Gretton et al., 2005b) or the Hilbert-Schmidt independence criterion (HSIC) (Gretton et al., 2005a). It has been noted that BiDO-HSIC has better defense performance than BiDO-COCO. We choose BiDO-HSIC and use $\lambda_x$ and $\lambda_y$ to control $d(z, x)$ and $d(z, y)$ separately.

**NLS (Struppek et al., 2023).** Negative Label Smoothing (NLS) converts hard labels into soft labels by incorporating a negative smoothing factor $\lambda$ into the cross-entropy loss, which affects the optimization process in MIAs.

**TLDMI (Ho et al., 2024).** TLDMI utilizes a model pretrained on public datasets and transfers the earlier layers to the target model. The transferred layers are frozen, and only the later layers are trained on the private dataset. According to their analysis, the earlier layers are more vulnerable to MIAs; thus, TLDMI helps reduce the private information encoded in the model. We define the ratio of transferred and frozen parameters as the hyperparameter $\lambda$.

**Differential Privacy (Abadi et al., 2016).** Differential Privacy (DP) was initially introduced to provide privacy guarantees for algorithms operating on aggregate databases (Dwork, 2006; Dwork et al., 2014). It was later adapted to deep learning through Differentially Private Stochastic Gradient Descent (DP-SGD) (Abadi et al., 2016). To limit the influence of a single sample on model updates and prevent excessive information leakage, DP-SGD first computes the $l_2$-norm of each sample's gradient and clips it if it exceeds a predefined threshold. We set this threshold to 1. To further enhance privacy, even if an attacker gains access to the model parameters, they should not be able to accurately infer specific training samples. To achieve this, random perturbations drawn from Gaussian noise are added during gradient updates. We define the noise ratio as the hyperparameter $\lambda$.

**NoPeek (Vepakomma et al., 2020).** NoPeek is a widely used defense method in CI that measures and reduces the correlation between intermediate features and the input, thereby preventing server adversaries from reconstructing the original input data. The loss function for this method is defined as follows:

$$\mathcal{L} = \alpha \cdot DCOR(X_{priv}, f_C(X_{priv}))$$
$$+ (1 - \alpha) \cdot TASK(Y_{priv}, f_S(f_C(X_{priv}))) \tag{11}$$

where $DCOR$ represents the distance correlation metric, and $TASK$ denotes the classification loss between the true label and the model's prediction. By jointly minimizing this loss, a better trade-off can be achieved between preserving input data privacy and maintaining model utility.

**Noise (Titcombe et al., 2021).** Titcombe et al. (Titcombe et al., 2021) proposed a defense approach that adds Laplacian noise directly to the intermediate features before transmission to the server, aiming to hinder input reconstruction. This added randomness increases the difficulty for adversaries to infer the mapping between the intermediate features and the original input. In our implementation, we set the noise mean to 0 and control its variance using $\lambda$.

**Dropout (He et al., 2020).** Dropout randomly disables a subset of neurons during the forward pass, stochastically altering the activation patterns of intermediate representations. We control the dropout probability using $\lambda$.

**DISCO (Singh et al., 2021).** DISCO learns a dynamic, data-driven pruning filter to selectively obfuscate sensitive information in the feature space. It monitors an attacker and learns the optimal pruning strategy to defend against it. We set the pruning rate to $\lambda$ and balance the main task loss and the monitored adversarial loss with weights of 0.5 each.

**InfoScissors (Duan et al., 2024).** InfoScissors reduces the mutual information between a model's intermediate features and both the input and predictions. Since predictions are held by the server in this paper, we focus solely on minimizing the mutual information between the intermediate features and the input. The mutual information loss is controlled by the parameter $\alpha$.

# D DATASETS

**CelebA (different ID).** We select another 2,504 images from 234 different identities for this dataset, with no overlap with the private CelebA dataset used in the main experiments.

**AI-Face (Lin et al., 2025)** is the first million-scale, demographically annotated AI-generated face image dataset, including real faces, deepfake video frames, and faces generated by 10 GANs and 8 diffusion models. We select 20,000 images generated by Latent Diffusion (Rombach et al., 2022). We choose intersectional classification (gender and skin tone) with 6 classes: 0-(Female, Light), 1-(Female, Medium), 2-(Female, Dark), 3-(Male, Light), 4-(Male, Medium), 5-(Male, Dark).

*Hyperparameters.* For the main experiments and the aforementioned face dataset, we use the same hyperparameters. All the images are resized to $224 \times 224$. For training the target models, we use the Adam optimizer with a batch size of 128 and a learning rate of 0.001. The models are trained for 100 epochs on CelebA and 50 epochs on FaceScrub. For Step1, we use a batch size of 64, a learning rate of 0.001, and train for 100 epochs. For Step2 and r-MTIA, we fine-tune the recovered model with a batch size of 64, a learning rate of 0.0001, and only 5 epochs.

**Neurotechnology UareU** (Neurotechnology, 2007) is a fingerprint dataset distributed by Neurotechnology, containing 65 fingers, each with 8 impressions.

**FVC2004** (Maltoni et al., 2009; of Bologna), 2003) is a fingerprint dataset introduced in the Third International Fingerprint Verification Competition. It consists of four databases, with a total of 40 fingers, each having 8 impressions.

*Hyperparameters.* For fingerprint recognition, we use Neurotechnology UareU as $D_{priv}$ and FVC2004 as $D_{pub}$. The images are resized to $224 \times 224$ and trained on the MobileNetV2 model. For the target model, we train for 20 epochs with a batch size of 8 and a learning rate of 0.001. For MTIA Step 1, we train for 70 epochs with a batch size of 8 and a learning rate of 0.001. In MTIA Step 2, we train for 5 epochs with a batch size of 8 and a learning rate of 0.0001.

**PCE-SynthPalm-1.6M** (Jin et al., 2024; PCE-SynthPalm-1.6M, 2024) is a synthetic palmprint dataset designed to address the lack of large-scale datasets in palmprint recognition research. It includes 1.6 million palmprint images spanning 50,000 subjects.

*Hyperparameters.* For palmprint recognition, we select 50 identities from the PCE-SynthPalm-1.6M dataset as $D_{priv}$ and another 100 identities as $D_{pub}$. The images are resized to $224 \times 224$ and trained on the MobileNetV2 model. For the target model, we train for 20 epochs with a batch size of 32 and a learning rate of 0.001. For MTIA Step 1, we train for 30 epochs with a batch size of 32 and a learning rate of 0.001. In MTIA Step 2, we train for 5 epochs with a batch size of 32 and a learning rate of 0.0001.

**Imagenette (Howard, 2019)** is a curated subset of ImageNet (Deng et al., 2009), consisting of 10 easily distinguishable classes: tench, English springer, cassette player, chain saw, church, French horn, garbage truck, gas pump, golf ball, and parachute. It contains a total of 13,394 images.

**Imagewoof (Howard, 2019)** is a subset of ImageNet (Deng et al., 2009) comprising 10 dog breeds that are more challenging to classify due to their visual similarity. It contains a total of 12,954 images. The included breeds are: Australian Terrier, Border Terrier, Samoyed, Beagle, Shih Tzu, English Foxhound, Rhodesian Ridgeback, Dingo, Golden Retriever, and Old English Sheepdog.

*Hyperparameters.* For both datasets, the images are resized to $224 \times 224$ and trained on the MobileNetV2 model. For the target model, we train for 20 epochs with a batch size of 128 and a

learning rate of 0.001. For MTIA Step 1, we train for 50 epochs with a batch size of 128 and a learning rate of 0.001. In MTIA Step 2, we train for 5 epochs with a batch size of 128 and a learning rate of 0.0001.

# E  MODELS

Both MobileNetV2 and ResNet-50 are composed of three parts: an initial convolutional layer, multiple blocks, and a final classification layer. The initial convolutional layer processes the input image and maps the three RGB channels into multiple feature channels. In MobileNetV2, it consists of a $3{\times}3$ convolution, a batch normalization layer, and a ReLU activation. In ResNet-50, it consists of a $7{\times}7$ convolution, batch normalization, a ReLU activation, and a max-pooling layer. In our main experimental setup, the initial convolutional layer and the first block (an inverted residual block for MobileNetV2 and a residual block for ResNet-50) are deployed on the client, while the remaining blocks are deployed on the server. When the split point moves deeper, additional blocks are shifted to the client. For the attacker, a VGG block is used as a substitute for the client models. The architectures of the VGG block, inverted residual block, and residual block are shown in Figure 10.

The VGG block consists of two convolutional layers with a kernel size of $3 \times 3$. Each convolutional layer is followed by a batch normalization layer and a ReLU activation function. At the end of the VGG block, a max-pooling layer is used for dimensionality reduction.

The inverted residual block in MobileNetV2 consists of convolutional layers with two different kernel sizes: $1 \times 1$ and $3 \times 3$. Specifically, it follows a dimension expansion-first approach, followed by dimension reduction. It first applies a $1 \times 1$ pointwise convolution to expand the number of input channels, followed by a $3 \times 3$ depthwise separable convolution to extract features. Finally, another $1 \times 1$ pointwise convolution compresses the number of channels back to the original count. The inverted residual block uses ReLU6 as the activation function, which restricts the output values between 0 and 6. Additionally, a skip connection is employed to add the inputs and outputs of the block.

The residual block in ResNet-50 also consists of convolutional layers with two different kernel sizes but differs from the inverted residual block. It employs a $1 \times 1$ convolution for dimension reduction, followed by a $3{\times}3$ convolution for feature extraction, and finally another $1{\times}1$ convolution to expand the dimensions. This structure, known as the bottleneck design, features larger dimensions at both ends and a smaller dimension in the middle, effectively reducing computational complexity. The residual block includes a skip connection that directly adds the input to the output passing through two convolutional layers, facilitating gradient propagation. Unlike the inverted residual block, a ReLU activation function is applied after the skip connection.

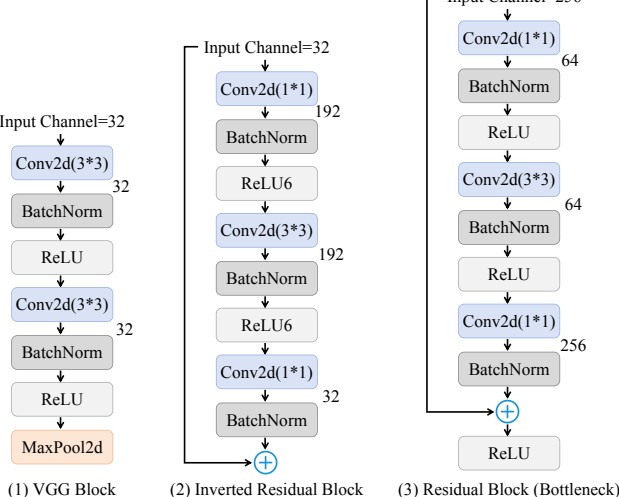

Figure 10: Architectures of three different blocks.

# F ADDITIONAL EXPERIMENTS

## F.1 ADDITIONAL DEFENSES

We evaluate our attack against two stronger defense mechanisms under various hyperparameters: Noisy_ARL (Jeong et al., 2023) and CEM (Xia et al., 2025). Since CEM needs to be combined with other defenses, we pair it with NoPeek (Vepakomma et al., 2020), Dropout (He et al., 2020), and Noisy_ARL (Jeong et al., 2023). The results are shown in Table 6. MTIA remains effective against both defenses, improving the recovered model accuracy from 57.03% to 75.06% and the inversion attack accuracy from 59.33% to 79.53%.

Table 6: Attack results against additional defenses.

| Defense | Hyperparams | Method | Test-Acc ↑ | PPA Att-Acc ↑ |
|---------|-------------|--------|------------|---------------|
| w/o | - | Target | 88.15 | 90.93±1.3 |
| | | MTIA | 77.05 | 72.93±2.2 |
| Norsy_ARL | (2, 0.01) | Target | 88.09 | 92.00±0.2 |
| | | MTIA | 75.06 | 79.53±2.2 |
| | (5, 0.01) | Target | 83.76 | 82.26±2.0 |
| | | MTIA | 68.94 | 72.93±1.2 |
| | (10, 0.01) | Target | 82.59 | 71.60±1.1 |
| | | MTIA | 64.90 | 67.26±3.4 |
| NoPeek_CEM | (0.01, 1, 0.5) | Target | 83.57 | 85.73±1.1 |
| | | MTIA | 66.73 | 71.53±1.6 |
| | (0.01, 1, 0.7) | Target | 80.93 | 82.53±1.3 |
| | | MTIA | 57.03 | 59.33±2.1 |
| Dropout_CEM | (0.01, 1, 0.3) | Target | 84.71 | 88.13±1.9 |
| | | MTIA | 68.32 | 69.73±1.4 |
| | (0.01, 1, 0.5) | Target | 81.39 | 86.73±2.0 |
| | | MTIA | 71.48 | 71.20±0.9 |
| Norsy_ARL_CEM | (1.0, 10, 0.01) | Target | 85.29 | 86.19±1.1 |
| | | MTIA | 65.91 | 68.80±1.6 |
| | (5.0, 10, 0.01) | Target | 83.34 | 81.06±1.7 |
| | | MTIA | 64.25 | 71.33±1.3 |

## F.2 COMPUTATIONAL OVERHEAD

We calculate the computational overhead of each step to better illustrate the attack process, as shown in Table 7. Training details can be found in Appendix C and Appendix D. For MTIA, Step 1 completes the model weights through transfer learning, Step 2 fine-tunes the model via self-attention alignment, and the inversion attack reconstructs the images. r-MTIA adds a fine-tuning process using the reconstructed images and a second inversion (if needed). Step 1 requires more time than training the target model, while Step 2 incurs only a small computational cost. The inversion step is time-consuming because the latent space of the GAN must be optimized hundreds of times for each identity. For r-MTIA, the fine-tuning cost is low, and most of the computation is spent on inversion. Therefore, repeating the attack more than once is unnecessary due to its high computational cost, as the reconstructed results already achieve high accuracy and additional repetitions yield only diminishing gains.

Table 7: Computational overhead (GPU hours) of different processes: MTIA involves Step 1, Step 2, and inversion, while r-MTIA involves Step 1, Step 2, inversion, fine-tuning, and a second inversion.

| Dataset | Model | Target Model Training | Attack Step | | | | Total Attack Time | |
|---|---|---|---|---|---|---|---|---|
| | | | Step1 | Step2 | Inversion | Fine-tuning | MTIA | r-MTIA |
| CelebA | MobileNetV2 | 7.68 | 11.21 | 0.09 | 19.92 | 0.38 | 31.22 | 51.52 |
| | ResNet-50 | 8.46 | 12.46 | 0.13 | 23.57 | 0.39 | 36.16 | 60.12 |
| FaceScrub | MobileNetV2 | 5.44 | 8.80 | 0.09 | 10.55 | 0.22 | 19.44 | 30.21 |
| | ResNet-50 | 5.74 | 9.14 | 0.12 | 12.65 | 0.22 | 21.91 | 34.78 |

### F.3 SELF-ATTENTION ALIGNMENT LAYERS

To evaluate the impact of using different layers for self-attention alignment, we experiment with four portions of the early layers: 100%, 50%, 30%, and 10%. These portions indicate the number of layers counted from the first layer relative to the total number of layers. The recovered model accuracy is shown in Figure 11. Using fewer layers for alignment slightly reduces performance and slows convergence, while also lowering the computational cost.

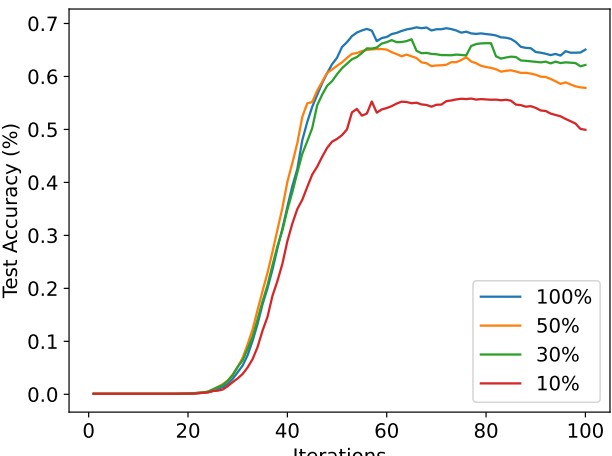

Figure 11: Test accuracy of the recovered model using different portions of self-attention alignment layers.

### F.4 Overlapping Analysis between CelebA and FaceScrub

To analyze the overlapping IDs between CelebA and FaceScrub, we trained MobileNetV2 and ResNet-50 models on each dataset, resulting in four models in total. We then performed cross verification by feeding CelebA images into the FaceScrub-trained models and vice versa, and recorded the predicted labels for each ID. For each ID, we identified the most frequently predicted label and computed its proportion among all predictions for that ID as the match ratio. We calculated the average match ratio across the four models and report the results in Figure 12. IDs with a match ratio above 0.5 were identified as overlapping, yielding 22 such cases. The images of these overlapping IDs are shown in Table 8. The identified 22 IDs indeed correspond to the same individuals, while those with match ratios below 0.5 are visually similar but not the same person.

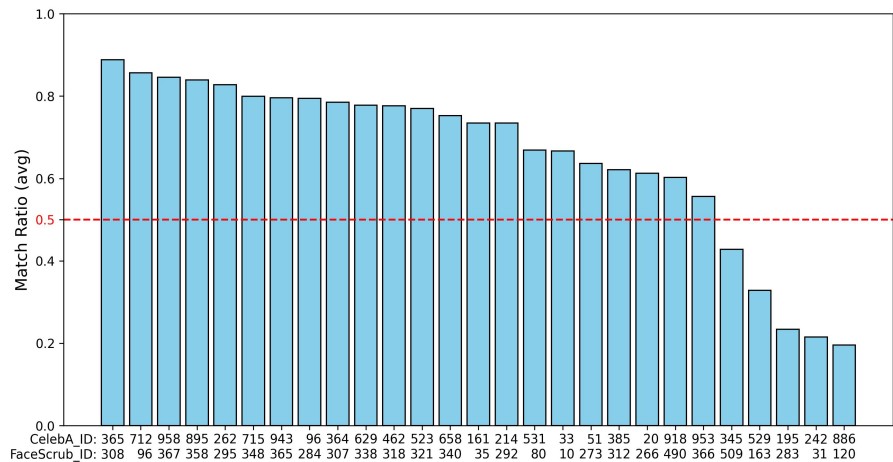

Figure 12: Average match ratio for overlaped predictions of four models.

Table 8: The corresponding image of the overlaped IDs between CelebA and FaceScrub.

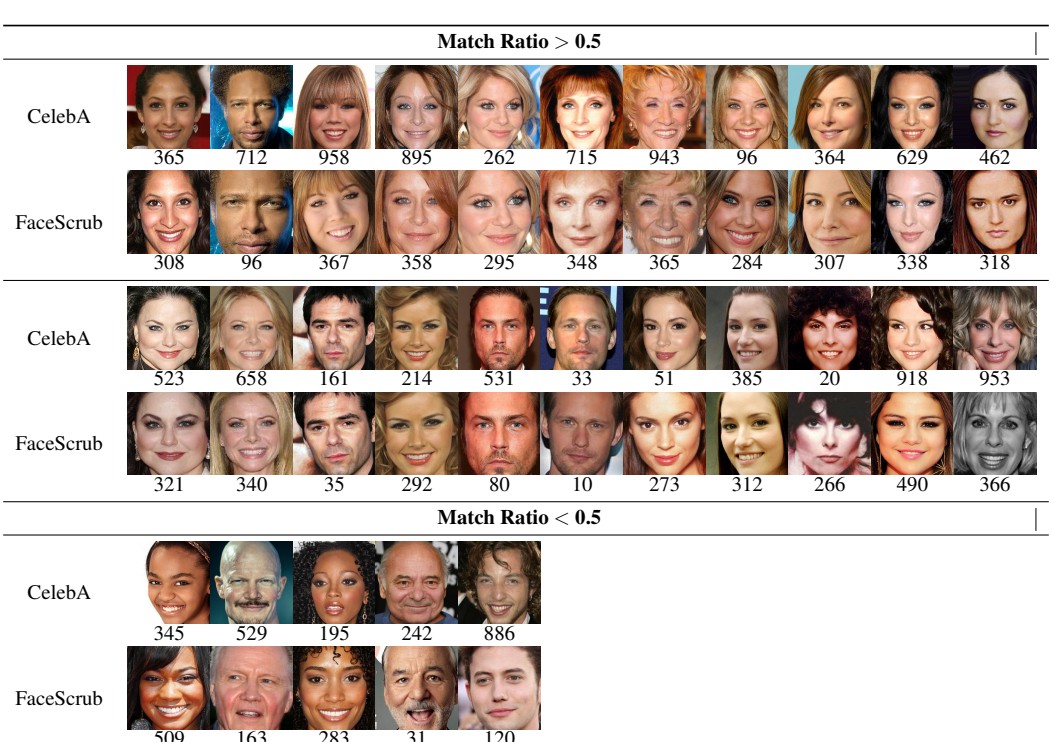

## F.5 ATTACK RESULTS USING DIFFERENT CELEBA IDENTITY

The overlapping IDs are few, only 2.2% in CelebA and 4.1% in FaceScrub, and they are associated with different labels. This label mismatch can disrupt classification during our attack, as images of the same identity are assigned different labels. To fully eliminate this overlap, we conduct two new experiments. Our target model is trained on CelebA using 1000 IDs, and we select another 234 CelebA IDs as our public dataset for attacks, forming a non-overlapping set. We also remove the 22 overlapping IDs identified in the FaceScrub dataset in Section F.4 and re-evaluate on this cleaned dataset. The results are shown in Table 9. MTIA remains effective on both new public datasets, successfully restoring model functionality and reconstructing images. Compared to the original results on the full FaceScrub dataset reported in the main paper, the performance decreases only slightly—from 77.05% to 76.80%/74.78% for MobileNetV2 and from 71.39% to 61.45%/71.22% for ResNet-50.

Table 9: Attack results using different datasets. ↑ and ↓ indicate that higher and lower scores, respectively, correspond to better attack performance.

| $D_{priv}$ | $D_{pub}$ | Method | MobileNetV2 | | | ResNet-50 | | |
|---|---|---|---|---|---|---|---|---|
| | | | TestAcc ↑ | PPA AttAcc ↑ | FDist ↓ | TestAcc ↑ | PPA AttAcc ↑ | FDist ↓ |
| CelebA | CelebA (Different 234 ID) | Target | 88.15 | 90.93±1.3 | 122.56 | 87.67 | 92.33±0.8 | 150.23 |
| | | Pretrain | 0.03 | 0.13±0.1 | 278.92 | 0.09 | 0.06±0.1 | 299.69 |
| | | Pretrain-SA | 11.26 | 13.86±1.7 | 218.47 | 2.34 | 2.53±0.7 | 250.48 |
| | | TL | 0.13 | 0.86±0.3 | 260.46 | 0.68 | 26.26±1.7 | 242.60 |
| | | MTIA | **76.80** | **73.60±2.5** | **144.66** | **61.45** | **71.73±0.6** | **168.10** |
| | FaceScrub (22 Overlap Removed) | Target | 88.15 | 90.93±1.3 | 122.56 | 87.67 | 92.33±0.8 | 150.23 |
| | | Pretrain | 0.01 | 0.13±0.1 | 275.66 | 0.09 | 0.13±0.1 | 296.66 |
| | | Pretrain-SA | 4.03 | 4.60±1.4 | 245.61 | 3.71 | 3.33±0.5 | 249.16 |
| | | TL | 0.48 | 2.93±0.8 | 254.32 | 29.98 | 53.33±1.6 | 209.91 |
| | | MTIA | **74.78** | **71.26±1.5** | **148.26** | **71.22** | **75.13±2.0** | **172.19** |

### F.6 ANALYSIS OF LAYER ATTENTION

We visualize the attention maps of the MobileNetV2 model trained on CelebA using Grad-CAM Selvaraju et al. (2017), from the first layer to the last, as shown in Figure 13. The first two maps in the top row correspond to the client model. For the target model, shallow layers focus on fine-grained details such as hair and nose, while deeper layers attend to broader, less detailed regions. Attention patterns between neighboring layers are similar and show smooth, continuous transitions. With TL, the early-layer attention becomes inconsistent with the originals, and the inter-layer relations become less coherent, causing deviations that grow cumulatively with depth. In contrast, self-attention alignment preserves continuity across layers, enabling more accurate feature extraction.

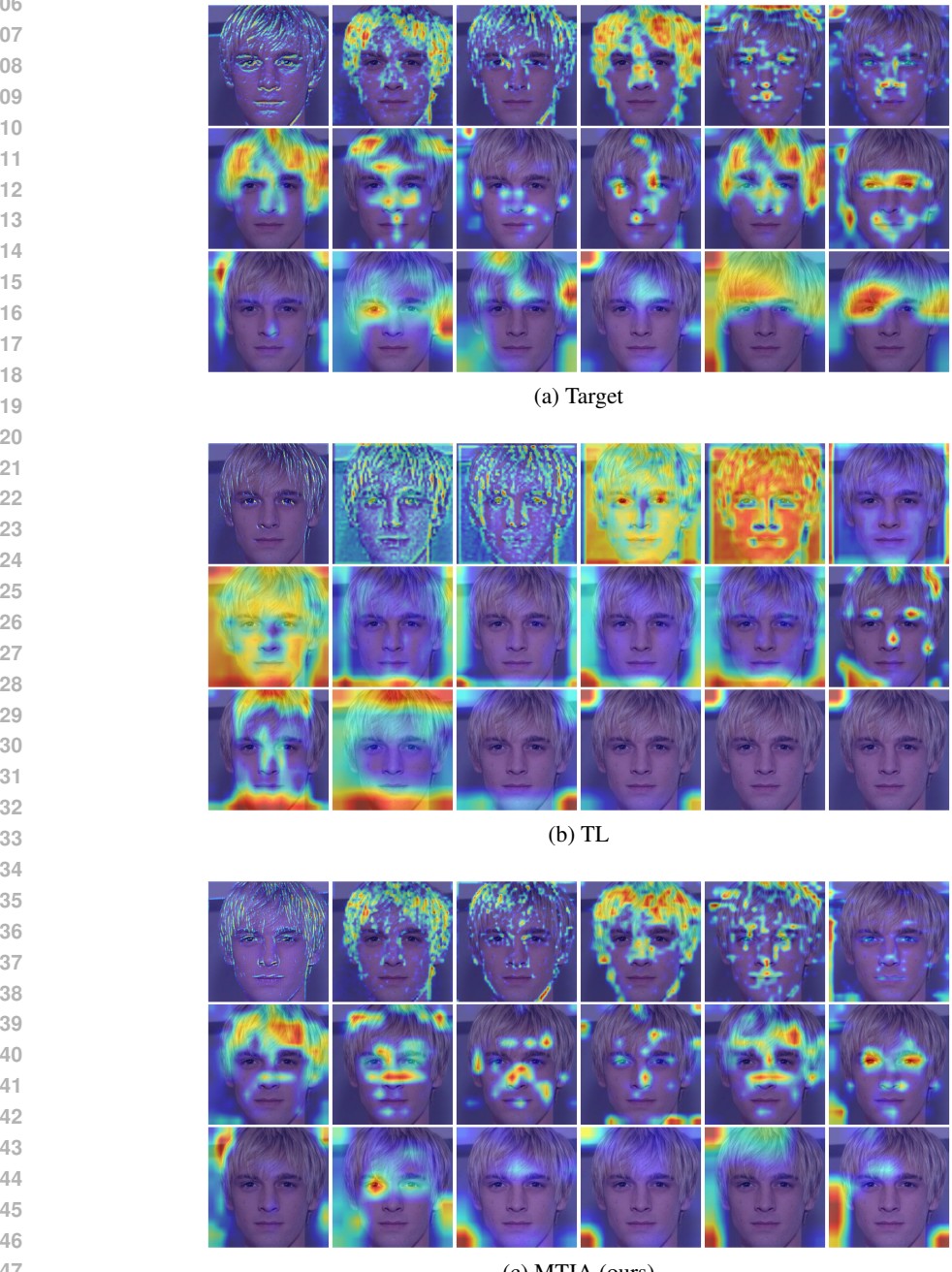

(a) Target

(b) TL

(c) MTIA (ours)

Figure 13: Attention maps of MobileNetV2 trained on CelebA, from the first layer to the last layer.

## F.7 Fine-tuning of r-MTIA

The repeated MTIA (r-MTIA) uses reconstructed images to fine-tune the recovered model and enhance its classification performance. Since the reconstructed images may be distorted or imperfect and may not resemble the real identities, several fine-tuning strategies are employed to mitigate the risk of error accumulation as follows:

- *None*: The model is directly fine-tuned using all reconstructed images.
- *Image Filtering*: We adopt the same methods as in PPA (Struppek et al., 2022), applying random image transformations (such as RandomResizedCrop and RandomHorizontalFlip) to the reconstructed images. Images that still receive high prediction scores after transformations are selected as the final fine-tuning dataset. This step filters out adversarial examples and low-similarity images.
- *Layer Freezing*: Parameters belonging to the original server model are frozen, and only the recovered part is fine-tuned.
- *L2-SP*: An L2 penalty at the starting point (L2-SP) (Xuhong et al., 2018) is applied during fine-tuning to prevent the parameters from deviating excessively from their initial values. Denoting the recovered model parameters as $\theta_r$, the penalty is computed as: $l_{sp} = ||\theta_r^* - \theta_r||_2^2$.

The experimental results are shown in Table 10. Simply using all reconstructed images for fine-tuning can slightly enhance model performance. Applying image filtering further improves performance on CelebA, while the improvement on FaceScrub is less pronounced. This may be because the quality of reconstructed images for FaceScrub is higher than for CelebA, making filtering imperfect images more beneficial for CelebA. Layer Freezing and L2-SP also provide notable improvements for CelebA but are less effective for FaceScrub.

Table 10: Accuracy of different fine-tuning methods by r-MTIA.

| Dataset | Model | MTIA Accuracy | r-MTIA Accuracy | | | |
|---|---|---|---|---|---|---|
| | | | None | Image Filtering | Image Filtering + Layer Freezing | Image Filtering + L2-SP |
| CelebA | MobileNetV2 | 77.05 | 78.91 (+1.86) | 79.62 (+2.57) | **80.54** (+3.49) | 79.43 (+2.38) |
| | ResNet-50 | 68.32 | 69.17 (+0.85) | **71.38** (+3.06) | 70.73 (+2.41) | 71.02 (+2.70) |
| FaceScrub | MobileNetV2 | 88.37 | 90.05 (+1.68) | **90.07** (+1.70) | 89.72 (+1.35) | 89.72 (+1.35) |
| | ResNet-50 | 85.93 | **87.32** (+1.39) | 86.55 (+0.62) | 85.56 (-0.37) | 85.84 (-0.09) |

## F.8 ATTACK RESULTS WITH QUERY ACCESS

If the attacker has query access to the client model, they can recover it through a distillation-based approach. We consider two types of distillation methods as follows:

- *Feature Distillation (FD)*: The attacker queries the client model with the public dataset $(X_{pub})$ to obtain the intermediate features $(f_C(X_{pub}))$ and minimizes the discrepancy between these features and those produced by the substitute client $(\hat{f}_C(X_{pub}))$. The loss is computed as:

$$\mathcal{L}_{FD} = \left\| f_C(X_{pub}) - \hat{f}_C(X_{pub}) \right\|_2^2 \tag{12}$$

- *Knowledge Distillation (KD)*: The attacker queries both the client model and the server model with the public dataset $(X_{pub})$ to obtain the final outputs $(f_S(f_C(X_{pub})))$. The attacker then minimizes the Kullback–Leibler divergence (KL) Csiszár (1975) between these outputs and those produced by the substitute client and server, $(f_S(\hat{f}_C(X_{pub})))$. This loss, commonly known as Knowledge Distillation (KD) Hinton et al. (2015), is computed as:

$$y_C = \text{Softmax}(f_S(f_C(X_{pub}))) \tag{13}$$

$$\hat{y}_C = \text{Softmax}(f_S(\hat{f}_C(X_{pub}))) \tag{14}$$

$$\mathcal{L}_{KD} = KL(y_C || \hat{y}_C) \tag{15}$$

We evaluate the model-recovery performance of FD and KD in Table 11. Given query access, an adversary can submit an unlimited number of inputs to the client model and obtain either the intermediate features or the final model outputs. With these signals, the adversary is able to reconstruct the entire model with high fidelity, resulting in a highly accurate recovery.

Table 11: Attack results with query access.

| Dataset | Method | TestAcc ↑ | |
| --- | --- | --- | --- |
| | | MobileNetV2 | ResNet-50 |
| CelebA | Target | 88.15 | 87.67 |
| | FD | 74.77 | 84.53 |
| | KD | 76.98 | 84.01 |
| FaceScrub | Target | 93.48 | 93.96 |
| | FD | 88.09 | 93.02 |
| | KD | 89.59 | 92.84 |

### F.9 OTHER ADDITIONAL RESULTS

Additional identity inversion results are presented in Figure 14.

Additional LRC scores on FaceScrub are shown in Table 12.

Additional feature visualization by t-SNE is shown in Fig 15, 16 and 17.

Results of MTIA on MaxViT (Tu et al., 2022) model are shown in Table 13.

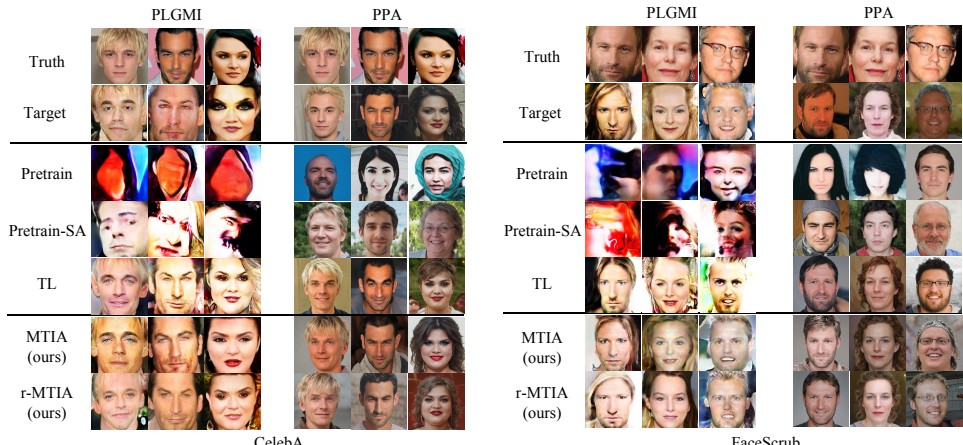

Figure 14: Identity inversion results on ResNet-50. "Truth" refers to the ground truth images of the target identity. "Target" refers to the reconstructed images of the whole target model.

Table 12: LRC score on FaceScrub.

| Method | MobileNetV2 | ResNet-50 |
|---|---|---|
| Pretrain | -0.144 | -0.056 |
| Pretrain-SA | 0.289 | 0.012 |
| TL | -0.046 | 0.609 |
| MTIA (ours) | **0.862** | **0.758** |

Table 13: Attack results on MaxViT.

| Dataset | Method | Test-Acc ↑ | PPA Att-Acc ↑ | PPA Feat-Dist ↓ |
|---|---|---|---|---|
| CelebA | Target | 87.54 | 71.79±1.8 | 143.77 |
| CelebA | MTIA (ours) | 72.65 | 59.66±2.8 | 156.26 |
| FaceScrub | Target | 94.75 | 83.33±2.4 | 130.59 |
| FaceScrub | MTIA (ours) | 85.75 | 66.79±3.1 | 142.62 |

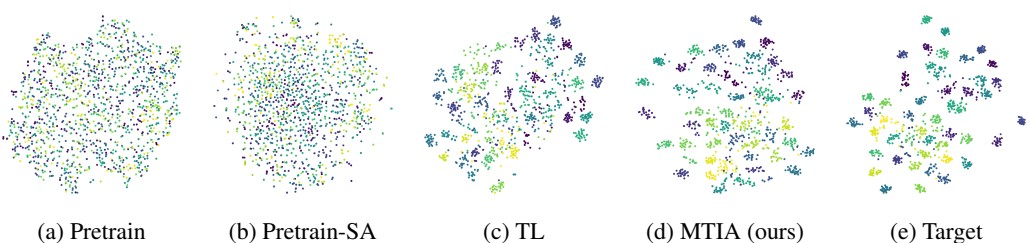

| (a) Pretrain | (b) Pretrain-SA | (c) TL | (d) MTIA (ours) | (e) Target |
|---|---|---|---|---|

Figure 15: Feature visualization by t-SNE on CelebA and ResNet-50.

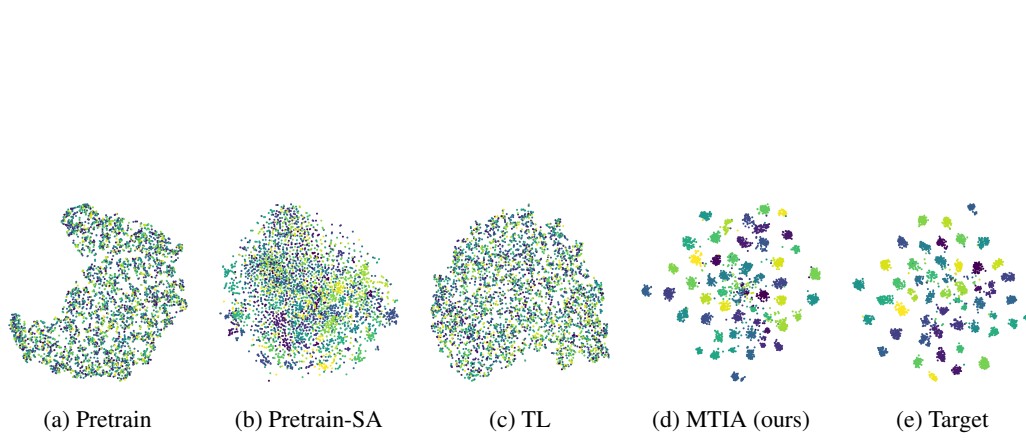

(a) Pretrain      (b) Pretrain-SA      (c) TL      (d) MTIA (ours)      (e) Target

Figure 16: Feature visualization by t-SNE on FaceScrub and MobileNetV2.

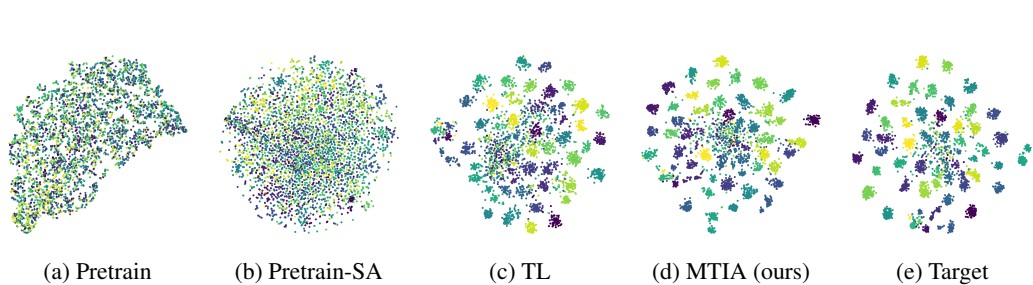

(a) Pretrain      (b) Pretrain-SA      (c) TL      (d) MTIA (ours)      (e) Target

Figure 17: Feature visualization by t-SNE on FaceScrub and ResNet-50.

