# OpenReview forum: "Model Theft and Inversion Attacks Against Query-free Collaborative Inference Systems"
_ICLR.cc/2026/Conference — Submitted to ICLR 2026_

### Official Review · Reviewer_UEvU · 2025-10-27

**Soundness:** 2
**Presentation:** 3
**Contribution:** 2
**Rating:** 4
**Confidence:** 3

**Summary:**

This paper addresses privacy vulnerabilities in collaborative inference systems, where models are split between a client and a server. It introduces a novel threat, Model Theft and Inversion Attacks (MTIA), under a challenging and realistic scenario: the server-side adversary is query-free and only possesses a public dataset that is label-inconsistent with the client's private data. The proposed MTIA framework operates in two stages: (1) a transfer-learning-based model completion step to substitute the missing client-side model, and (2) a self-attention alignment step that uses the public dataset to align the substitute model's feature space with the server's model. Experimental results demonstrate that MTIA can successfully recover the client model's functionality with high fidelity, which subsequently enables high-fidelity model inversion attacks to reconstruct sensitive private training data.

**Strengths:**

1. The paper is well-written, presenting a clear and logical flow from problem definition and methodology to experimental validation.

2. The primary contribution is the introduction of a novel and practical threat model (query-free and label-inconsistent).

3. The paper proposes an effective two-stage framework with transfer learning and self-attention to solve this new problem scenario.

**Weaknesses:**

1. This paper assumes a collaborative inference scenario where only a single block is assigned to the client side. In other words, it is assumed that most of the model, excluding one block, is already known in the case of the ResNet-50 or MobileNetV2 used in the experiments. The additional experiments presented in Figure 4, which evaluate up to three blocks, also show similar results, where the server side holds the majority of the model. If the model is evenly partitioned or the client holds more blocks, failing to validate MTIA in these scenarios could suggest that the model is only applicable in situations where sufficient information is already available.

2. There is almost no performance difference between the pretrained model and the Transfer Learning (TL) only setting in the main paper, while there is a significant performance gap between TL and MTIA. This indicates that the alignment step is responsible for the majority of the performance gain. However, further explanation is needed as to why aligning attention maps using a public dataset helps in aligning the functionality of the model to its private dataset.

3. Apart from the problem formulation and framework design, the other components lack novelty. Model inversion attack and attention alignment are both well-established methods in the literature.

4. Despite model inversion being a central component of the proposed MTIA threat, the paper’s review of related work is notably incomplete. The review is narrowly focused on specific white-box, GAN-based methods (PPA and PLGMI) that are operationally required for the paper’s experiments. It fails to cite foundational works that established the model inversion threat (e.g., Fredrikson et al. [1]) as well as subsequent research from other attack settings, such as black-box attacks (RLB-MI [2]) and label-only attacks (BREP-MI [3]).

[1 ] Fredrikson, M., Jha, S., & Ristenpart, T. (2015, October). Model inversion attacks that exploit confidence information and basic countermeasures. In Proceedings of the 22nd ACM SIGSAC conference on computer and communications security (pp. 1322-1333).

[2] Han, G., Choi, J., Lee, H., & Kim, J. (2023). Reinforcement learning-based black-box model inversion attacks. In Proceedings of the IEEE/CVF Conference on Computer Vision and Pattern Recognition (pp. 20504-20513).

[3] Kahla, M., Chen, S., Just, H. A., & Jia, R. (2022). Label-only model inversion attacks via boundary repulsion. In Proceedings of the IEEE/CVF conference on computer vision and pattern recognition (pp. 15045-15053).

**Questions:**

1. Your main experiments assume a server-heavy split where the client only has one block. Do you expect the MTIA attack to remain effective in more realistic scenarios, such as a 50:50 split or a client-heavy split where the server has less information?

2. The self-attention alignment step provides the most significant performance boost. Can you provide a more fundamental explanation for why aligning attention maps using a public, label-inconsistent dataset also works to align the model for the private task?

3. When you retrain the model on the reconstructed images, how do you account for the risk of error accumulation? Is it possible that training on these imperfect images could introduce noise and actually degrade the model's performance?

---

> ### Author Response · Authors · 2025-11-19
> **Response to Reviewer UEvU**
>
> We sincerely thank the reviewer for the valuable feedback and constructive suggestions. We have carefully addressed each point as detailed below, and we hope that our responses sufficiently clarify the concerns raised.
>
> We have uploaded a revised version that includes the following modifications in the Appendix C.2, F.6 and F.7. The modifications are highlighted in blue. Since we cannot exceed the page limit in the main paper, we will incorporate the important experiments in the Appendix into the main paper in the final version.
>
> **Weaknesses 1 & Questions 1 (Regarding splitting setting):**
>
> Thank you for pointing out this issue. In realistic scenarios and in the original design intention of collaborative inference, the client device is lightweight and resource‑constrained in both storage and RAM, making it unable to store large weights or run large models. The server, by contrast, has greater storage, more RAM, and hardware acceleration for faster processing. Therefore, the split point is typically shallow, with larger weights deployed on the server. Since our proposed MTIA already operates under minimal information: query‑free, label‑inconsistent, and partial‑model, conducting an attack in this setting is already extremely challenging. Thus, using a 50:50 split or a client-heavy split would make the attack even more difficult, as the server would have even less information to work with.
>
> **Weaknesses 2 & Questions 2 (Regarding explanation for self-attention alignment):**
>
> We visualize the attention maps of the MobileNetV2 model trained on CelebA using Grad-CAM [1], from the first layer to the last, as shown in Figure 13 of Appendix F.6. The first two maps in the top row correspond to the client model. For the target model, shallow layers focus on fine-grained details such as hair and nose, while deeper layers attend to broader, less detailed regions. Attention patterns between neighboring layers are similar and show smooth, continuous transitions. With TL, the early-layer attention becomes inconsistent with the originals, and the inter-layer relations become less coherent, causing deviations that grow cumulatively with depth. In contrast, self-attention alignment preserves continuity across layers, enabling more accurate feature extraction. This is also consistent with the original paper of SAD, using self-attention alignment improves feature extraction and preserves information for subsequent layers, preventing forgetting.
>
> [1] Ramprasaath R Selvaraju, Michael Cogswell, Abhishek Das, Ramakrishna Vedantam, Devi Parikh, and Dhruv Batra. Grad-cam: Visual explanations from deep networks via gradient-based localization. In Proceedings of the IEEE international conference on computer vision, pp. 618–626, 2017.
>
> **Weaknesses 3 (Regarding novelty):**
>
> Self-attention distillation was originally proposed to enhance a model’s representation learning without relying on teacher distillation, not for privacy analysis. We are the first to apply it for model recovery in a more realistic and challenging setting. Privacy risks under a query-free and label-inconsistent attacker have never been explored before, and existing methods fail entirely in this scenario. Analyzing attacks under these constraints is therefore more relevant to real-world applications.
>
> Our key insight is that the attention patterns between layers are similar, and the deeper layers can guide the shallow layers toward better information extraction and alignment, thereby enabling the theft of user privacy. The two-step model recovery fully exploits information encoded in the deeper layers and enables more severe reconstruction of personal images. We further provide a fine-tuning strategy to strengthen the attack. The overall attack design is tailored to the query-free setting and operates under significant limitations. MTIA thus represents a realistic and more severe threat with stronger practical applicability.
>
>
> **Weaknesses 4 (Regarding citation):**
>
> Thank you for pointing out this issue. We are sorry for missing some foundational works and alternative attack settings. Since our proposed attack already has access to the white-box model weights, we primarily cited the main white-box inversion attacks in the paper. To the best of our knowledge, we have now included all relevant model inversion attacks in Appendix C.2, including white-box, black-box, and label-only. We will incorporate this section into the main paper in the final version.

---

> ### Author Response · Authors · 2025-11-19
> **Response to Reviewer UEvU**
>
> **Questions 3 (Regarding error accumulation):**
>
> Thank you for pointing out this interesting direction. The r-MTIA uses reconstructed images to fine-tune the recovered model and improve its classification performance. The reconstructed images may be distorted or imperfect, and may not fully resemble the real identities, potentially degrading model performance. But our evaluation shows that the accumulated errors are outweighed by the gains provided by high-quality reconstructed samples. As a result, the model’s performance still improves.
>
> To further enhance performance, several fine-tuning strategies are employed to mitigate the risk of error accumulation, as follows:
>
> - *None*: The model is directly fine-tuned using all reconstructed images.
> - *Image Filtering*: We adopt the same methods as in PPA, applying random image transformations (such as RandomResizedCrop and RandomHorizontalFlip) to the reconstructed images. Images that still receive high prediction scores after transformations are selected as the final fine-tuning dataset. This step filters out adversarial examples and low-similarity images.
> - *Layer Freezing*: Parameters belonging to the original server model are frozen, and only the recovered part is fine-tuned.
> - *L2-SP*: An L2 penalty at the starting point (L2-SP) [2] is applied during fine-tuning to prevent the parameters from deviating excessively from their initial values. Denoting the recovered model parameters as $\theta_r$, the penalty is computed as: $l_{sp}=\left|| \theta_r^*-\theta_r \right||_2^2$.
>
> The experimental results are shown in the Table below, which is also included in Appendix F.7. Simply using all reconstructed images for fine-tuning can slightly enhance model performance. Applying image filtering further improves performance on CelebA, while the improvement on FaceScrub is less pronounced. This may be because the quality of reconstructed images for FaceScrub is higher than for CelebA, making filtering imperfect images more beneficial for CelebA. Layer Freezing and L2-SP also provide notable improvements for CelebA but are less effective for FaceScrub.
>
> | Dataset    | Model       | MTIA Accuracy | r-MTIA Accuracy | | | |
> |-----------|------------|---------------|-----|------|--------|-----------------|
> |           |            |               | None | Image Filtering | Image Filtering + Layer Freezing | Image Filtering + L2-SP |
> | CelebA    | MobileNetV2| 77.05         | 78.91 (+1.86)  | 79.62 (+2.57)  | **80.54 (+3.49)**  | 79.43 (+2.38) |
> |     | ResNet-50  | 68.32         | 69.17 (+0.85)  | **71.38 (+3.06)** | 70.73 (+2.41)     | 71.02 (+2.70) |
> | FaceScrub | MobileNetV2| 88.37         | 90.05 (+1.68)  | **90.07 (+1.70)** | 89.72 (+1.35)     | 89.72 (+1.35) |
> |  | ResNet-50  | 85.93         | **87.32 (+1.39)** | 86.55 (+0.62)  | 85.56 (-0.37)     | 85.84 (-0.09) |
>
> [2] LI Xuhong, Yves Grandvalet, and Franck Davoine. Explicit inductive bias for transfer learning
> with convolutional networks. In International conference on machine learning, pp. 2825–2834.
> PMLR, 2018.

---

> ### Author Response · Authors · 2025-11-27
> **Official Comment by Authors**
>
> Dear Reviewer UEvU,
>
> We sincerely appreciate your time and effort in reviewing our manuscript and providing valuable feedback.
>
> As the rebuttal period is nearing its end, we would like to kindly inquire whether our responses have adequately addressed your concerns. We have provided detailed clarifications and additional experimental results in our reply, and we would be more than happy to discuss any remaining questions you might have.
>
> Thank you again for your consideration.
>
> Best regards,
>
> The Authors

---

### Official Review · Reviewer_q8x5 · 2025-10-31

**Soundness:** 1
**Presentation:** 3
**Contribution:** 1
**Rating:** 2
**Confidence:** 3

**Summary:**

This paper introduces Model Theft and Inversion Attacks (MTIA) for recovering a missing client model by accessing the server system in collaborative inference. MTIA first reconstructs the client model using a self-attention guided, transfer-based approach. The recovered model is then used to apply existing model inversion attacks and reconstruct the client's training data, which is subsequently used to further refine the recovered model. Authors demonstrate the effectiveness of MTIA across two datasets and two model architectures.

**Strengths:**

- The results demonstrate that MTIA is highly effective, able to recover the client model with a high test accuracy of 79-90\%.

- MTIA is able to recover the client model without using any query to client model.

**Weaknesses:**

- The proposed method focuses solely on recovering the client model and does not introduce any new approach for model inversion attacks. In fact, the authors employ PPA and PLGMI in Phase 2 to recover training data from the newly obtained client model. Therefore, using the term “Model Theft and Inversion Attacks” appears to overstate the actual contribution of the paper.

- The experimental results are questionable. The authors use the FaceScrub and CelebA datasets, one being public and the other private. Although the Adversary’s Knowledge section claims that the public and private datasets are non-overlapping, this assumption may not hold true for FaceScrub and CelebA. To the best of my knowledge, several classes overlap between these two datasets, which could artificially boost the performance of the client model.

- The server architecture is not clearly described. For instance, the authors mention using MobileNetV2 and ResNet50, but it is unclear which layers belong to the original client model and which are part of the server model.

- The paper lacks formal analysis or theoretical justification to explain why using self-guided attention should lead to improved performance in the new client model

**Questions:**

- Please clarify the potential overlap between the public and private datasets.

- Please provide a clearer description of the server architecture.

- Could the authors provide analysis or explaining why using self-guided attention should lead to improved performance in the new client model?

---

> ### Author Response · Authors · 2025-11-19
> **Response to Reviewer q8x5**
>
> We sincerely thank the reviewer for the valuable feedback and constructive suggestions. We have carefully addressed each point as detailed below, and we hope that our responses sufficiently clarify the concerns raised.
>
> We have uploaded a revised version that includes the following modifications in the Appendix E, F.4, F.5 and F.6. The modifications are highlighted in blue. Since we cannot exceed the page limit in the main paper, we will incorporate the important experiments in the Appendix into the main paper in the final version.
>
> **Weaknesses 1 (Regarding contribution):**
>
> We refer to the attack as “Model Theft and Inversion Attacks” because it has two purposes: model leakage and data leakage. Both goals are important and jointly contribute to user privacy risks. While the primary component of MTIA focuses on recovering the client model, this recovery is an essential prerequisite for the subsequent inversion attack. Critically, the white-box inversion attack uses the stolen model to reconstruct private images with high fidelity, showing that the threat extends far beyond model theft and directly compromises user privacy. Together, these two components make the attacker significantly more threatening in realistic scenarios.

---

> ### Author Response · Authors · 2025-11-19
> **Response to Reviewer q8x5**
>
> **Weaknesses 2 & Questions 1 (Regarding datasets overlapping):**
>
> We chose CelebA and FaceScrub to create a more realistic setting, where the distributions differ significantly and the attacker’s capabilities are limited. However, the amount of overlap exceeded our expectations, so we conducted a thorough overlap analysis as described below.
>
> We trained MobileNetV2 and ResNet-50 models on each dataset, resulting in four models in total. We then performed cross verification by feeding CelebA images into the FaceScrub-trained models and vice versa, and recorded the predicted labels for each ID. For each ID, we identified the most frequently predicted label and computed its proportion among all predictions for that ID as the match ratio. We calculated the average match ratio across the four models and report the results in Figure 12 of Appendix F.4. IDs with a match ratio above 0.5 were identified as overlapping, yielding 22 such cases. The images of these overlapping IDs are shown in Table 8 of Appendix F.4. The identified 22 IDs indeed correspond to the same individuals, while those with match ratios below 0.5 are visually similar but not the same person.
>
> But the overlapping IDs are few, only 2.2% in CelebA and 4.1% in FaceScrub, and they are associated with different labels. This label mismatch can disrupt classification during our attack, as images of the same identity are assigned different labels. To fully eliminate this overlap, we conduct two new experiments. Our target model is trained on CelebA using 1000 IDs, and we select another 234 CelebA IDs as our public dataset for attacks, forming a non-overlapping set. We also remove the 22 overlapping IDs identified in the FaceScrub dataset and re-evaluate on this cleaned dataset. The results are shown in the Table below, which is also included in Appendix F.5. MTIA remains effective on both new public datasets, successfully restoring model functionality and reconstructing images. Compared to the original results on the full FaceScrub dataset reported in the main paper, the performance decreases only slightly—from 77.05% to 76.80%/74.78% for MobileNetV2 and from 71.39% to 61.45%/71.22% for ResNet-50.
>
> This indicates that our proposed MTIA remains effective on non-overlapping datasets, and the overlap has minimal impact and contributes little to the attack results.
>
> | D_priv | D_pub                          | Method      | MobileNetV2      |   |        | ResNet-50      |    |        |
> |--------|--------------------------------|-------------|-------------|--------|--------|------------|---------|-------|
> |        |                                |             | TestAcc ↑ | AttAcc ↑ | FDist ↓ | TestAcc ↑ | AttAcc ↑ | FDist ↓ |
> | CelebA | CelebA (Different 234 ID)      | Target      | 88.15     | 90.93 ± 1.3 | 122.56 | 87.67     | 92.33 ± 0.8 | 150.23 |
> |  |       | Pretrain    | 0.03      | 0.13 ± 0.1  | 278.92 | 0.09      | 0.06 ± 0.1  | 299.69 |
> |  |       | Pretrain-SA | 11.26     | 13.86 ± 1.7 | 218.47 | 2.34      | 2.53 ± 0.7  | 250.48 |
> |  |       | TL          | 0.13      | 0.86 ± 0.3  | 260.46 | 0.68      | 26.26 ± 1.7 | 242.60 |
> |  |       | MTIA        | **76.80**     | **73.60 ± 2.5** | **144.66** | **61.45**     | **71.73 ± 0.6** | **168.10** |
> | CelebA | FaceScrub (22 Overlap Removed) | Target      | 88.15     | 90.93 ± 1.3 | 122.56 | 87.67     | 92.33 ± 0.8 | 150.23 |
> |  |  | Pretrain    | 0.01      | 0.13 ± 0.1  | 275.66 | 0.09      | 0.13 ± 0.1  | 296.66 |
> |  |  | Pretrain-SA | 4.03      | 4.60 ± 1.4  | 245.61 | 3.71      | 3.33 ± 0.5  | 249.16 |
> |  |  | TL          | 0.48      | 2.93 ± 0.8  | 254.32 | 29.98     | 53.33±1.6 | 209.91 |
> |  |  | MTIA        | **74.78**     | **71.26 ± 1.5** | **148.26** | **71.22**     | **75.13 ± 2.0** | **172.19** |

---

> ### Author Response · Authors · 2025-11-19
> **Response to Reviewer q8x5**
>
> **Weaknesses 3 & Questions 2 (Regarding server architecture):**
>
> Both MobileNetV2 and ResNet-50 are composed of three parts: an initial convolutional layer, multiple blocks, and a final classification layer. The initial convolutional layer processes the input image and maps the three RGB channels into multiple feature channels. In MobileNetV2, it consists of a 3$\times$3 convolution, a batch normalization layer, and a ReLU activation. In ResNet-50, it consists of a 7$\times$7 convolution, batch normalization, a ReLU activation, and a max-pooling layer. In our main experimental setup, the initial convolutional layer and the first block (an inverted residual block for MobileNetV2 and a residual block for ResNet-50) are deployed on the client. The server holds the remaining blocks and the final classification layer. When the split point moves deeper, additional blocks are shifted to the client. For the server attacker, the initial and early blocks are missing and the images cannot be directly processed, so a VGG block is used as a substitute for the client models. We have included these modifications in Appendix E for clarity.
>
> **Weaknesses 4 & Questions 3 (Regarding explanation for self-attention alignment):**
>
> We include a visualization analysis to illustrate the underlying reason. We visualize the attention maps of the MobileNetV2 model trained on CelebA using Grad-CAM [1], from the first layer to the last, as shown in Figure 13 of Appendix F.6. The first two maps in the top row correspond to the client model. For the target model, shallow layers focus on fine-grained details such as hair and nose, while deeper layers attend to broader, less detailed regions. Attention patterns between neighboring layers are similar and show smooth, continuous transitions. With TL, the early-layer attention becomes inconsistent with the originals, and the inter-layer relations become less coherent, causing deviations that grow cumulatively with depth. In contrast, self-attention alignment preserves continuity across layers, enabling more accurate feature extraction. This is also consistent with the original paper of SAD, using self-attention alignment improves feature extraction and preserves information for subsequent layers, preventing forgetting.
>
> [1] Ramprasaath R Selvaraju, Michael Cogswell, Abhishek Das, Ramakrishna Vedantam, Devi Parikh, and Dhruv Batra. Grad-cam: Visual explanations from deep networks via gradient-based localization. In Proceedings of the IEEE international conference on computer vision, pp. 618–626, 2017.

---

> ### Author Response · Authors · 2025-11-27
> **Official Comment by Authors**
>
> Dear Reviewer q8x5,
>
> We sincerely appreciate your time and effort in reviewing our manuscript and providing valuable feedback.
>
> As the rebuttal period is nearing its end, we would like to kindly inquire whether our responses have adequately addressed your concerns. We have provided detailed clarifications and additional experimental results in our reply, and we would be more than happy to discuss any remaining questions you might have.
>
> Thank you again for your consideration.
>
> Best regards,
>
> The Authors

---

### Official Review · Reviewer_Bn6V · 2025-11-01

**Soundness:** 3
**Presentation:** 3
**Contribution:** 2
**Rating:** 6
**Confidence:** 3

**Summary:**

This paper presents a comprehensive study on Model Theft and Inversion Attacks (MTIA) against query-free collaborative inference systems. The proposed MTIA framework introduces a two-step strategy that combines transfer learning and self-attention alignment to recover client-side models and subsequently reconstruct private data under highly restrictive conditions (label inconsistency, no query access). The paper demonstrates strong performance on multiple datasets (CelebA, FaceScrub, fingerprints, and object classification) and also evaluates robustness against several existing defenses (NoPeek, BiDO, DP, etc.). The topic is timely and relevant to privacy and security in distributed ML systems.

**Strengths:**

1. The paper studies a realistic yet underexplored attack setting, query-free collaborative inference with label inconsistency. This fills a meaningful gap compared to prior works assuming either full query access or label-consistent auxiliary datasets.

2. The two-stage design (transfer-based recovery + self-attention alignment) is clearly described, mathematically formalized, and experimentally validated. The theoretical explanation of the self-attention alignment loss (Eq. 3–6) is technically rigorous and intuitive.

3. The experiments are well-organized, spanning diverse datasets and architectures, and the paper conducts extensive ablation studies (e.g., effect of missing blocks, dataset size, fine-tuning epochs). Evaluation against multiple attack/defense baselines is thorough.

**Weaknesses:**

1. While several defenses are tested (Tables 3–4), it would be valuable to analyze how MTIA behaves under stronger modern defense paradigms, such as: 1)Adversarially-trained encoders for privacy-preserving inference (e.g., Noisy Adversarial Representation Learning, UAI 2023 [1]); 2)Defense approaches offering theoretical robustness guarantees (e.g., Theoretical Insights in Model Inversion Robustness and Conditional Entropy Maximization for Collaborative Inference Systems, CVPR 2024 [2]) Including these could strengthen the discussion on the boundary of MTIA’s capability and clarify how robust the proposed method is against more principled protection schemes.

2. The repeated MTIA attack (r-MTIA) involves multiple fine-tuning and GAN optimization cycles. A brief analysis of computational overhead and scalability (e.g., GPU hours, convergence time) would make the results more practical and comparable.

3. It would be helpful to discuss how sensitive the self-attention alignment loss is to layer selection (Eq. 3–6), and whether aligning only selected layers could reduce computational cost without hurting performance.

**Questions:**

Please refer to the listed weaknesses. The reviewer is interested in seeing whether the proposed attack methods remain effective against stronger defense mechanisms, such as adversarial representation learning and defenses offering theoretical robustness guarantees.

---

> ### Author Response · Authors · 2025-11-19
> **Response to Reviewer Bn6V**
>
> We sincerely thank the reviewer for the valuable feedback and constructive suggestions. We have carefully addressed each point as detailed below, and we hope that our responses sufficiently clarify the concerns raised.
>
> We have uploaded a revised version that includes the following modifications in the Appendix F.1, F.2, and F.3. The modifications are highlighted in blue. Since we cannot exceed the page limit in the main paper, we will incorporate the important experiments in the Appendix into the main paper in the final version.
>
> **Weaknesses 1 & Questions (Regarding additional defenses):**
>
> We evaluate our attack against two stronger defense mechanisms under various hyperparameters: Noisy_ARL [1] and CEM [2]. Since CEM needs to be combined with other defenses, we pair it with NoPeek, Dropout, and Noisy_ARL. The results are shown in the Table below. MTIA remains effective against both defenses, improving the recovered model accuracy from 57.03% to 75.06% and the inversion attack accuracy from 59.33% to 79.53%. This demonstrates the effectiveness of our proposed attacks, as a model with only 57.03% accuracy can still leak private information. We also include these experiments in Appendix F.1 for clearer presentation.
>
> [1] Jonghu Jeong, Minyong Cho, Philipp Benz, and Tae-hoon Kim. Noisy adversarial representation learning for effective and efficient image obfuscation. In Uncertainty in Artificial Intelligence, pp. 953–962. PMLR, 2023.
>
> [2] Song Xia, Yi Yu, Wenhan Yang, Meiwen Ding, Zhuo Chen, Ling-Yu Duan, Alex C Kot, and Xudong Jiang. Theoretical insights in model inversion robustness and conditional entropy maximization for collaborative inference systems. In Proceedings of the Computer Vision and Pattern Recognition Conference, pp. 8753–8763, 2025.
>
> | Defense           | Hyperparams        | Method | Test-Acc ↑ | PPA Att-Acc ↑        |
> |-------------------|--------------------|--------|------------|-----------------------|
> | w/o               | -                  | Target | 88.15      | 90.93 ± 1.3           |
> |                   |                    | MTIA   | 77.05      | 72.93 ± 2.2           |
> | Norsy_ARL         | (2, 0.01)          | Target | 88.09      | 92.00 ± 0.2           |
> |                   |                    | MTIA   | 75.06      | 79.53 ± 2.2           |
> |                   | (5, 0.01)          | Target | 83.76      | 82.26 ± 2.0           |
> |                   |                    | MTIA   | 68.94      | 72.93 ± 1.2           |
> |                   | (10, 0.01)         | Target | 82.59      | 71.60 ± 1.1           |
> |                   |                    | MTIA   | 64.90      | 67.26 ± 3.4           |
> | NoPeek_CEM        | (0.01, 1, 0.5)     | Target | 83.57      | 85.73 ± 1.1           |
> |                   |                    | MTIA   | 66.73      | 71.53 ± 1.6           |
> |                   | (0.01, 1, 0.7)     | Target | 80.93      | 82.53 ± 1.3           |
> |                   |                    | MTIA   | 57.03      | 59.33 ± 2.1           |
> | Dropout_CEM       | (0.01, 1, 0.3)     | Target | 84.71      | 88.13 ± 1.9           |
> |                   |                    | MTIA   | 68.32      | 69.73 ± 1.4           |
> |                   | (0.01, 1, 0.5)     | Target | 81.39      | 86.73 ± 2.0           |
> |                   |                    | MTIA   | 71.48      | 71.20 ± 0.9           |
> | Norsy_ARL_CEM     | (1.0, 10, 0.01)    | Target | 85.29      | 86.19 ± 1.1           |
> |                   |                    | MTIA   | 65.91      | 68.80 ± 1.6           |
> |                   | (5.0, 10, 0.01)    | Target | 83.34      | 81.06 ± 1.7           |
> |                   |              | MTIA   |  64.25   | 71.33 ± 1.3               |

---

> ### Author Response · Authors · 2025-11-19
> **Response to Reviewer Bn6V**
>
> **Weaknesses 2 (Regarding computational overhead and scalability):**
>
> We provide a detailed breakdown of the computational overhead (GPU hours) for each step in the Table below, which is also included in Appendix F.2. The training details are the same as in Appendix C and Appendix D. For MTIA, Step 1 completes the model weights through transfer learning, Step 2 fine-tunes the model via self-attention alignment, and the inversion attack reconstructs the images. r-MTIA adds a fine-tuning process using the reconstructed images and a second inversion (if needed). Step 1 requires more time than training the target model, while Step 2 incurs only a small computational cost. The inversion step is time-consuming because the latent space of the GAN must be optimized hundreds of times for each identity. For r-MTIA, the fine-tuning cost is low, and most of the computation is spent on inversion. Therefore, repeating the attack more than once is unnecessary due to its high computational cost, as the reconstructed results already achieve high accuracy and additional repetitions yield only diminishing gains.
>
> | Dataset   | Model       | Target Model Training | Step1 | Step2 | Inversion | Fine-tuning | MTIA  | r-MTIA |
> | --------- | ----------- | --------------------- | ----- | ----- | --------- | ----------- | ----- | ------ |
> | CelebA    | MobileNetV2 | 7.68                  | 11.21 | 0.09  | 19.92     | 0.38        | 31.22 | 51.52  |
> |           | ResNet-50   | 8.46                  | 12.46 | 0.13  | 23.57     | 0.39        | 36.16 | 60.12  |
> | FaceScrub | MobileNetV2 | 5.44                  | 8.80  | 0.09  | 10.55     | 0.22        | 19.44 | 30.21  |
> |           | ResNet-50   | 5.74                  | 9.14  | 0.12  | 12.65     | 0.22        | 21.91 | 34.78  |
>
> **Weaknesses 3 (Regarding different layers for self-attention alignment loss):**
>
> To evaluate the impact of using different layers for self-attention alignment, we experiment with four portions of the early layers: 100%, 50%, 30%, and 10%. These portions indicate the number of layers counted from the first layer relative to the total number of layers. The recovered model accuracy is shown in Figure 11 in the Appendix F.3. Using fewer layers for alignment slightly reduces performance and slows convergence, while also lowering the computational cost. Therefore, leveraging all layers for self-attention alignment yields the best performance.

---

> > ### Comment · Reviewer_Bn6V · 2025-11-25
> > **Response to the authors**
> >
> > Thanks for your detailed response. My major concerns are solved.

---

> > > ### Author Response · Authors · 2025-11-26
> > > **Response to Reviewer Bn6V**
> > >
> > > Thank you for your positive feedback. We are glad that your major concerns have been resolved, and we appreciate your comments, which have helped improve our paper.

---

### Official Review · Reviewer_dQ4s · 2025-11-02

**Soundness:** 3
**Presentation:** 3
**Contribution:** 2
**Rating:** 6
**Confidence:** 3

**Summary:**

The paper studies a model-inversion attack against a targeted classification model  M in a query-free setting. The attacker does not know the model’s early feature extractor  F_c  but knows the final layers F_s. The attacker also possesses a rough approximation of the training data (public dataset  D_pub). The attack is “query-free” in the sense that the attacker cannot probe  F_c.
The goals are
- (G1) to recover a model F'  that approximates (F_c, F_s); and
- (G2) to recover representative instances for each class.

Specifically, the paper first trains a model F' with some layers frozen as F_s using the public dataset (goal G1).  Next employs known inversion techniques on F' to obtain (G2).

**Strengths:**

- The paper demonstrates that query-free model inversion is feasible when the attacker has a public dataset and access to the target model’s final layers.

- The manuscript is generally well written and clear. I did have some difficulty following Step 2 in Section 3.2, but the overall presentation is accessible.

- There is a broad literature on model inversion under varied threat models; the specific setting studied here, query-free attacks where only the last few layers are known, is technically interesting and could be a meaningful addition.

**Weaknesses:**

- The claim that the query-free setting is the more realistic threat model is not fully justified. In the described threat scenario the attacker is hosting F_s  and receiving features from clients; it is unclear why the attacker would be constrained to a query-free approach rather than using client-provided instances directly. The paper should better motivate why an attacker in this position would prefer or be forced into the query-free strategy.

**Questions:**

- Strengthen the threat-model discussion: explicitly compare the practical capabilities and trade-offs between a query-free attacker and "non-query-free" methods.

---

> ### Author Response · Authors · 2025-11-19
> **Response to Reviewer dQ4s**
>
> We sincerely thank the reviewer for the valuable feedback and constructive suggestions. We have carefully addressed each point as detailed below, and we hope that our responses sufficiently clarify the concerns raised.
>
> Weaknesses & Questions:
>
> For the scenario setting, we aim to analyze a privacy issue that is closer to real-world conditions. In this setting, the client model is deployed within an offline company, while the attacker is an external server not owned by the company. Consequently, the client-provided instances are not controllable by the server, which may lead to situations where no instances are sent, such as when the client device is offline or shut down. This is why we describe the setting as query-free, meaning the attacker cannot freely issue queries and can only passively wait for incoming instances.
>
> For the use of client-provided instances, the attacker does not have access to the original private inputs, which makes the received intermediate features difficult to exploit. In non-query-free methods, the attacker can send auxiliary data to the client and obtain corresponding intermediate features to learn the feature–input mapping or steal the client model through feature distillation. Therefore, in the non-query-free setting, the attacker can access substantially more information. In the query-free setting, the available information is limited, and consistent with real-world conditions, the instances may also be insufficient.
>
> We have uploaded a revised version that includes the above modifications in Appendix B. The modifications are highlighted in blue. Since we cannot exceed the page limit in the main paper, we will incorporate Appendix B into Section 3.1 for improved clarity in the final version.

---

> > ### Comment · Reviewer_dQ4s · 2025-11-25
> >
> > Thanks for the response, in particular on why  the attacker is unable to utilise the input instances.
> >
> > Since this being an "attack" work,  I agree that we should consider an attacker will limited information. Hence I'm not particularly concern on the lack of  analysis where the  attackers have more information. Nonetheless, for better insight, it would be helpful to see how much the attacker could gain by having the additional information.

---

> > > ### Author Response · Authors · 2025-11-26
> > > **Response to Reviewer dQ4s**
> > >
> > > Thank you for your comment. To assess how much additional advantage an attacker gains from having query access to the client model, we further examine the scenario in which the attacker performs model recovery through distillation-based methods. Specifically, if the attacker can query the client model, they can leverage these outputs to train a substitute model. We consider two types of distillation approaches, described as follows:
> > >
> > >
> > > - *Feature Distillation (FD)*: The attacker queries the client model with the public dataset ($X_{pub}$) to obtain the intermediate features ($f_C(X_{pub})$) and minimizes the discrepancy between these features and those produced by the substitute client ($\hat{f_C}(X_{pub})$). The loss is computed as:
> > >
> > >   $L_{FD} = || f_C(X_{pub}) - \hat{f_C}(X_{pub}) ||^2_2$
> > >
> > > - *Knowledge Distillation (KD)*: The attacker queries both the client model and the server model with the public dataset ($X_{pub}$) to obtain the final outputs ($f_S(f_C(X_{pub}))$). The attacker then minimizes the Kullback–Leibler divergence (KL) [1] between these outputs and those produced by the substitute client and server ($f_S(\hat{f_C}(X_{pub}))$). This loss, commonly known as Knowledge Distillation (KD) [2], is computed as:
> > >
> > >   $y_C = Softmax(f_S(f_C(X_{pub})))$
> > >
> > >   $\hat{y_C}= Softmax(f_S(\hat{f_C}(X_{pub})))$
> > >
> > >   $L_{KD} = KL(y_C || \hat{y}_C)$
> > >
> > > We evaluate the model-recovery performance of FD and KD in the Table below, which is also included in Appendix F.8. Given query access, an adversary can submit an unlimited number of inputs to the client model and obtain either the intermediate features or the final model outputs. With these signals, the adversary is able to reconstruct the entire model with high fidelity, resulting in a highly accurate recovery.
> > >
> > > [1] Imre Csisz´ar. I-divergence geometry of probability distributions and minimization problems. The annals of probability, pp. 146–158, 1975.
> > >
> > > [2] Geoffrey Hinton, Oriol Vinyals, and Jeff Dean. Distilling the knowledge in a neural network. arXiv preprint arXiv:1503.02531, 2015.
> > >
> > > | **Dataset** | **Method** | **TestAcc ↑ (MobileNetV2)** | **TestAcc ↑ (ResNet-50)** |
> > > |-------------|------------|------------------------------|-----------------------------|
> > > | **CelebA**  | Target     | 88.15                        | 87.67                       |
> > > |             | FD         | 74.77                        | 84.53                       |
> > > |             | KD         | 76.98                        | 84.01                       |
> > > | **FaceScrub** | Target   | 93.48                        | 93.96                       |
> > > |             | FD         | 88.09                        | 93.02                       |
> > > |             | KD         | 89.59                        | 92.84                       |

---

### Author Response · Authors · 2025-12-01
**Summary and General Response**

Dear ACs, SACs, and PCs,

We sincerely thank you for your time and effort on this work. We also thank the reviewers for their valuable feedback and constructive suggestions, which have improved our paper. We have carefully addressed each point in the rebuttal and made revisions in the updated PDF.

In this work, we propose a new model theft and inversion attack that reflects a more restricted and realistic scenario. The adversary is query-free, has access only to label-inconsistent datasets, and possesses only later partial models. The adversary performs a two-stage model recovery process based on knowledge transfer and self-attention alignment to fully restore the model’s accuracy, and then applies a model inversion attack to extract sensitive training data. The reconstructed data can further be used to improve the restored model’s accuracy.

Here, we provide a summary of the revisions:

1. Regarding the **setting**, we provide a more detailed explanation and conduct a new experiment where the adversary has query access, showing how much the attacker could gain from this additional information. (**Reviewer dQ4s, revisions in Appendix B and F.8**)

2. Regarding **additional defenses**, we conduct new experiments on more recent methods to show that our attack remains effective even under stronger defenses. (**Reviewer Bn6V, revisions in Appendix F.1**)

3. Regarding **computational overhead**, we provide a detailed breakdown of the GPU hours required for each step. (**Reviewer Bn6V, revisions in Appendix F.2**)

4. Regarding **the choice of layers for the self-attention alignment loss**, we conduct experiments using different portions of the early layers to show that including more layers leads to a stronger attack. (**Reviewer Bn6V, revisions in Appendix F.3**)

5. Regarding **dataset overlap**, we conduct a detailed overlap analysis and identify the overlapping IDs. We find that the overlap ratio is very small, and we further perform experiments using two non-overlapping datasets. The results show that overlap has nearly no impact on the attack, and we can achieve strong performance even with non-overlapping datasets. (**Reviewer q8x5, revisions in Appendix F.4 and F.5**)

6. Regarding the **server architecture**, we provide additional explanation of the splitting setting. (**Reviewer q8x5 and Reviewer UEvU, revisions in Appendix E**)

7. Regarding **the explanation of self-attention alignment**, we include a visualization analysis to illustrate the underlying rationale. It shows that attention patterns between neighboring layers are similar and exhibit smooth, continuous transitions, which benefit our attack. (**Reviewer q8x5 and Reviewer UEvU, revisions in Appendix F.6**)

8. Regarding **citations**, we have now included all relevant model inversion attacks to the best of our knowledge. (**Reviewer UEvU, revisions in Appendix C.2**)

9. Regarding **error accumulation**, we conduct new experiments using different fine-tuning strategies to further improve the recovered model’s accuracy. (**Reviewer UEvU, revisions in Appendix F.7**)

10. Regarding **contribution and novelty**, we provide additional explanation of our work. Our study addresses **two key goals: model leakage and data leakage**. Both are critical and together pose significant risks to user privacy. The primary model recovery in MTIA is an essential prerequisite for the subsequent inversion attack, which demonstrates that the threat extends beyond model theft and directly compromises user privacy. These two components combined make the attacker substantially more threatening **in realistic scenarios**. We are the first to apply this approach for model recovery in a more realistic and challenging setting. Privacy risks under a **query-free, label-inconsistent** attacker have never been explored before, and existing methods fail entirely in this scenario. Analyzing attacks under these constraints is therefore highly relevant to real-world applications. (**Reviewer q8x5 and Reviewer UEvU**)

We hope these clarifications, along with the demonstrated effectiveness of our method, warrant a positive reassessment of our work.

Best regards,

The Authors

---

### Meta-Review · Area_Chair_5WHm · 2026-01-10

**Summary:**

The reviewers in general had some concerns about the attack model and its practicality, i.e., query-free constraints on the server side.  There are also some serious concerns about the experimental setup and the lack of theoretical analysis of the attack model.

The authors did engage with the reviewers during the discussion and have provided some reasonable answers to most of the comments. But it’s not clear if some of the most critical reviewers (q8x5 and UEvU) would be satisfied with the answers. For example, the concern about the overlap of the FaceScrub and CelebA datasets was a valid one, and the authors post analyses and realization of this issue may not have eased the reviewer’s concerns.

Given the divergence of the ratings from the four reviewers and the lack of enthusiasm from the other two reviewers (both 6 – marginal accept), I would recommend a reject for this submission as is.

**Reviewer Concerns:**

The reviewers raised some interesting comments about this work, but not as many – this may in part indicate the lack of enthusiasm from the reviewers, or the lack of familiarity of the field (all with 3 confidence level).

For the questions raised by the reviewers, all have been addressed to some extend by the authors, such as
•	The setting on the client side (single block vs more blocks than server’s side) is not clear
•	The attack model is not justified (query free)
•	The computational cost
•	The concerns on the experimental setup (the overlap of the FaceScrub and CelebA dataset)

But their answers don’t seem to generate enough enthusiasm from reviewers

**Reviewer Scores:**

It's unlikely anyone would change their ratings.

---

### Decision · Program_Chairs · 2026-01-26

Reject